# The Peptidisc, a simple method for stabilizing membrane proteins in detergent-free solution

Michael Luke Carlson[1], John William Young[1], Zhiyu Zhao[1], Lucien Fabre[1], Daniel Jun[2,3], Jianing Li[4], Jun Li[4], Harveer Singh Dhupar[1], Irvin Wason[1], Allan T Mills[1], J Thomas Beatty[3], John S Klassen[4], Isabelle Rouiller[2], Franck Duong[1]*

[1]Department of Biochemistry and Molecular Biology, Faculty of Medicine, Life Sciences Institute, University of British Columbia, Vancouver, Canada; [2]Department of Anatomy and Cell Biology, McGill University, Montreal, Canada; [3]Department of Microbiology and Immunology, University of British Columbia, Vancouver, Canada; [4]Glycomics Centre and Department of Chemistry, University of Alberta, Alberta, Canada

**Abstract** Membrane proteins are difficult to work with due to their insolubility in aqueous solution and quite often their poor stability in detergent micelles. Here, we present the peptidisc for their facile capture into water-soluble particles. Unlike the nanodisc, which requires scaffold proteins of different lengths and precise amounts of matching lipids, reconstitution of detergent solubilized proteins in peptidisc only requires a short amphipathic bi-helical peptide ($NSP_r$) and no extra lipids. Multiple copies of the peptide wrap around to shield the membrane-exposed part of the target protein. We demonstrate the effectiveness of this 'one size fits all' method using five different membrane protein assemblies ($MalFGK_2$, FhuA, SecYEG, OmpF, BRC) during 'on-column', 'in-gel', and 'on-bead' reconstitution embedded within the membrane protein purification protocol. The peptidisc method is rapid and cost-effective, and it may emerge as a universal tool for high-throughput stabilization of membrane proteins to advance modern biological studies.
DOI: https://doi.org/10.7554/eLife.34085.001

*For correspondence:
fduong@mail.ubc.ca

## Introduction

Membrane proteins play essential roles, such as membrane transport, signal transduction, cell homeostasis, and energy metabolism. Despite their importance, obtaining these proteins in a stable non-aggregated state remains problematic. Membrane proteins are generally purified in detergent micelles, but these small amphipathic molecules are quite often detrimental to protein structure and activity, in addition to interfering with downstream analytical methods. This drawback has led researchers to develop detergent-free alternatives such as amphipols (*Popot, 2010*), SMALPs (*Lee et al., 2016*), saposin-lipoparticles (*Frauenfeld et al., 2016*), and the popular nanodisc system (*Bayburt et al., 2006*; *Denisov et al., 2004*). In the latter case, two amphipathic membrane scaffold proteins (MSPs) derived from apoA1 wrap around a small patch of lipid bilayer containing the target membrane protein (*Bayburt et al., 2006*; *Denisov et al., 2004*; *Denisov and Sligar, 2016*). However, in spite of an apparent simplicity, the formation of a nanodisc depends on several factors such as lipid to protein ratio, scaffold length, nature of lipids, rate of detergent removal and overall amenability of the target for re-assembly into lipid bilayer (*Bayburt et al., 2006*; *Denisov et al., 2004*; *Hagn et al., 2013*). The method is therefore not trivial and small deviations

**eLife digest** Surrounding every living cell is a biological membrane that is largely impermeable to water-soluble molecules. This hydrophobic (or "water-hating") barrier preserves the contents of the cell and also regulates how the cell interacts with its environment. This latter function is critical and relies on a class of proteins that are embedded within the membrane and are also hydrophobic.

The hydrophobic nature of membrane proteins is however inconvenient for biochemical studies which usually take place in water-based solutions. Therefore, membrane proteins are under-represented in biological research compared to the water-soluble ones, even though roughly one quarter of a cell's proteins are membrane proteins. Researchers have developed a few tricks to keep membrane proteins soluble after they have been extracted from the membrane. An old but popular technique makes use of detergents, which are chemicals with opposing hydrophobic and hydrophilic properties (hydrophilic literally means "water-loving"). However, even mild detergents can damage membrane proteins and will sometimes lead to experimental artifacts. More recent tricks to stabilize membrane proteins without detergents have been described but remain laborious, costly or difficult to perform.

To overcome these limitations, Carlson et al. developed a simple method to stabilize membrane proteins without detergent. Called the "peptidisc", the method uses multiple copies of a unique peptide – a short sequence of the building blocks of protein – that had been redesigned to have optimal hydrophobic and hydrophilic properties. The idea was that the peptides would wrap around the hydrophobic parts of the membrane protein, and shield them from the watery solution. Indeed, when Carlson et al. mixed this peptide with five different membrane proteins from bacteria, all were perfectly soluble and functional without detergent. The ideal ratio of peptide needed to form a peptidisc around each membrane protein was reached automatically, without having to test many different conditions. This indicates that the peptidisc acts like a "one size fits all" scaffold.

The peptidisc is a new tool that will allow more researchers, including those who are not expert biochemists, to study membrane proteins. This will yield a better understanding of the structure of a cell's membrane and how it interacts with the environment. Since the approach is both simple and easy to apply, more membrane proteins can now also be included in high-throughput searches for potential new drugs for various medical conditions.

DOI: https://doi.org/10.7554/eLife.34085.002

from optimal conditions often leads to low-efficiency reconstitution, or else liposome formation or protein aggregation (*Bayburt et al., 2006*).

Peptides have been considered as an alternative to scaffold proteins. Peptergents (*Corin et al., 2011*), lipopeptides (*Tao et al., 2013*), nanostructured [beta]-sheet peptides (*Privé, 2009*), and bi-helical derivatives of the ApoA1-mimetic 18A peptide, termed 'beltides' (*Larsen et al., 2016*), have been reported in the recent years. Yet, these systems have not been widely adopted for structural or functional characterization of membrane proteins for various reasons. Peptergents and lipopeptides can solubilize membrane proteins directly from a lipid bilayer, but same as detergent these peptides form mixed micelles that readily aggregate and precipitate below a certain critical micellar concentration (*Corin et al., 2011*; *Tao et al., 2013*). Beltides were shown to trap bacteriorhodopsin in solution, but the method required prior incubation with specific amounts of lipids and the particles formed were reportedly unstable at physiological temperatures (*Larsen et al., 2016*). Cost and complexity of the peptide can also be problematic. Nanostructured [beta]-sheet peptides contain extended alkyl chains covalently linked to glycine residues, while lipopeptides need to be covalently linked to a lipid molecule (*Tao et al., 2013*; *Privé, 2009*). These hydrophobic peptides are also difficult to work with given their low solubility. Peptergents require titration of base to become soluble in aqueous solution (*Corin et al., 2011*), and [beta]-sheet peptides can form extended filament clusters without detergent (*Privé, 2009*). Thus, a peptide-based reconstitution method that is cost-effective, rapid, unhindered by issues of solubility and generally applicable to membrane proteins remains to be developped.

We present the peptidisc. The peptidisc is made by multiple copies of an amphipathic bi-helical peptide (hereafter termed NSP$_r$) wrapping around its target membrane protein. In contrast to the

nanodisc, no additional lipids are necessary during reconstitution, except those that have co-purified with the protein (i.e. annular lipids). The peptide design we employ is a reverse version of the original nanodisc scaffold peptide (NSP), which consists of two repeats of the ApoA1-derived 18A peptide joined by a flexible linker proline (*Kariyazono et al., 2016*; *Chung et al., 1985*), in addition to two leucine residues which are substituted by phenylalanines to increase lipid affinity (*Kariyazono et al., 2016*; *Mishra et al., 2008*) (*Supplementary file 1*). The original NSP can stabilize lipid particles (*Kariyazono et al., 2016*), but issues of water-solubility and adaptability to membrane proteins were not demonstrated. We show here the NSP$_r$ design is very effective for stabilizing both α-helical and β-barrel membrane proteins of different size, topology, and complexity.

## Results

The original NSP is able to capture lipids from a lipid bilayer and forms discoidal particles of varying size (*Kariyazono et al., 2016*). However, due to its hydrophobicity, the NSP peptide is difficult to solubilize; the solution is cloudy after resuspension in water (*Figure 1B*). To increase peptide solubility, we reversed the amino acid sequence of NSP, so that the two amphipathic helices have a slightly altered orientation, leading to a lower hydrophobic moment and higher overall electropotential (*Figure 1AC*, *Supplementary file 1*). We also forewent acetylation and amidation of the peptide termini so that modifications such biotinylation remain possible. These small modifications allowed the final design (termed NSP$_r$) to be fully dissolved in water at concentrations up to 25 mg/ml (*Figure 1B*).

We tested the ability of NSP$_r$ to capture the ABC transporter MalFGK$_2$ using an 'on-column' reconstitution method (*Figure 2*). The NSP$_r$ peptide was mixed with MalFGK$_2$ in dodecyl maltoside and the mixture applied immediately onto a size exclusion column equilibrated in a detergent-free buffer (*Figure 2A*). The collected particles (hereafter termed peptidisc or MalFGK$_2$-NSP$_r$) were soluble and monodisperse, as shown by clear-native CN-PAGE and blue-native BN-PAGE (*Figure 2B*). We determined the approximate peptide content using SDS-PAGE (*Figure 3A*). Also, since annular lipids are tightly bound to membrane proteins (*Bechara et al., 2015*), we determined the final lipid content in the peptidisc using thin layer chromatography and photocolorimetric methods (*Figure 4A and B*). This analysis indicated a stoichiometry of 10 ± 2 peptides and 41 ± 10 lipids per MalFGK$_2$ (*Table 1*). This stoichiometry allowed us to calculate the mass of the MalFGK$_2$ peptidisc to 251 ± 12 kDa (*Table 1*). Interestingly, the lipids identified in the TLC analysis were predominantly negative phospholipids, cardiolipin and phosphatidylglycerol (*Figure 4B*). These lipids play a role in regulation of MalFGK$_2$ by stabilizing interactions with the regulatory protein EIIA (*Bao and Duong, 2013*). To corroborate the peptide and lipid stoichiometry, we determined the molecular weight of the intact complex by native mass spectrometry (247 ± 24 kDa; *Table 1*, *Figure 5A and B*), and size exclusion chromatography coupled multi-angle light scattering (SEC-MALS) (250 ± 17 kDa; *Table 2*, *Figure 6A*). The SEC-MALS analysis showed that the peptidisc remains perfectly stable during storage (e.g. 3 days at 4°C). We then examined the particles by single particle negative-stain electron microscopy (*Figure 2C*). The 2D-class averages revealed a structure very similar to MalFGK$_2$ in nanodiscs, (*Fabre et al., 2017*) with distinctly visible elements such as the MalK$_2$ dimer, the periplasmic P2 loop and a larger discoidal density corresponding to the NSP$_r$ peptides wrapping around the MalFG membrane domain. The measured diameter of the peptidisc was 11.7 ± 1.4 nm, which is consistent with a stoichiometry of 12 ± 2 peptides per MalFGK$_2$ complex when arranged in the double-belt model (*Table 2*). Finally, the ATPase activity of MalFGK$_2$ in peptidisc was similar to that reported in proteoliposomes and in nanodiscs (*Bao and Duong, 2012*), in sharp contrast to the high and unregulated ATPase activity observed in detergent micelles (*Figure 2D*). Importantly, the structural integrity of the peptidisc remained stable at elevated temperatures, with ~80% of MalFGK$_2$ peptidisc intact after incubation for 3 hr at 30°C (*Figure 6B*).

We also incorporated β-barrel membrane proteins in peptidisc, using the FhuA receptor as a model protein (*Figure 7A*). Analysis of FhuA-peptidisc by native mass spectrometry indicated a molecular weight of 138 ± 17 kDa (*Figure 5E and F*). Quantitation of the individual peptide and lipid components, following the approach applied to MalFGK$_2$ above, indicated an average of 8 ± 3 lipids and 10 ± 2 NSP$_r$ per FhuA (*Table 1*, *Figure 3B* and *Figure 4A*). These measurements were used to estimate a molecular mass of 131 ± 9 kDa (*Table 1*). Binding analysis on CN-PAGE further showed that FhuA in peptidisc is functional for both TonB and colicin M (*Figure 7B*). The binding of TonB

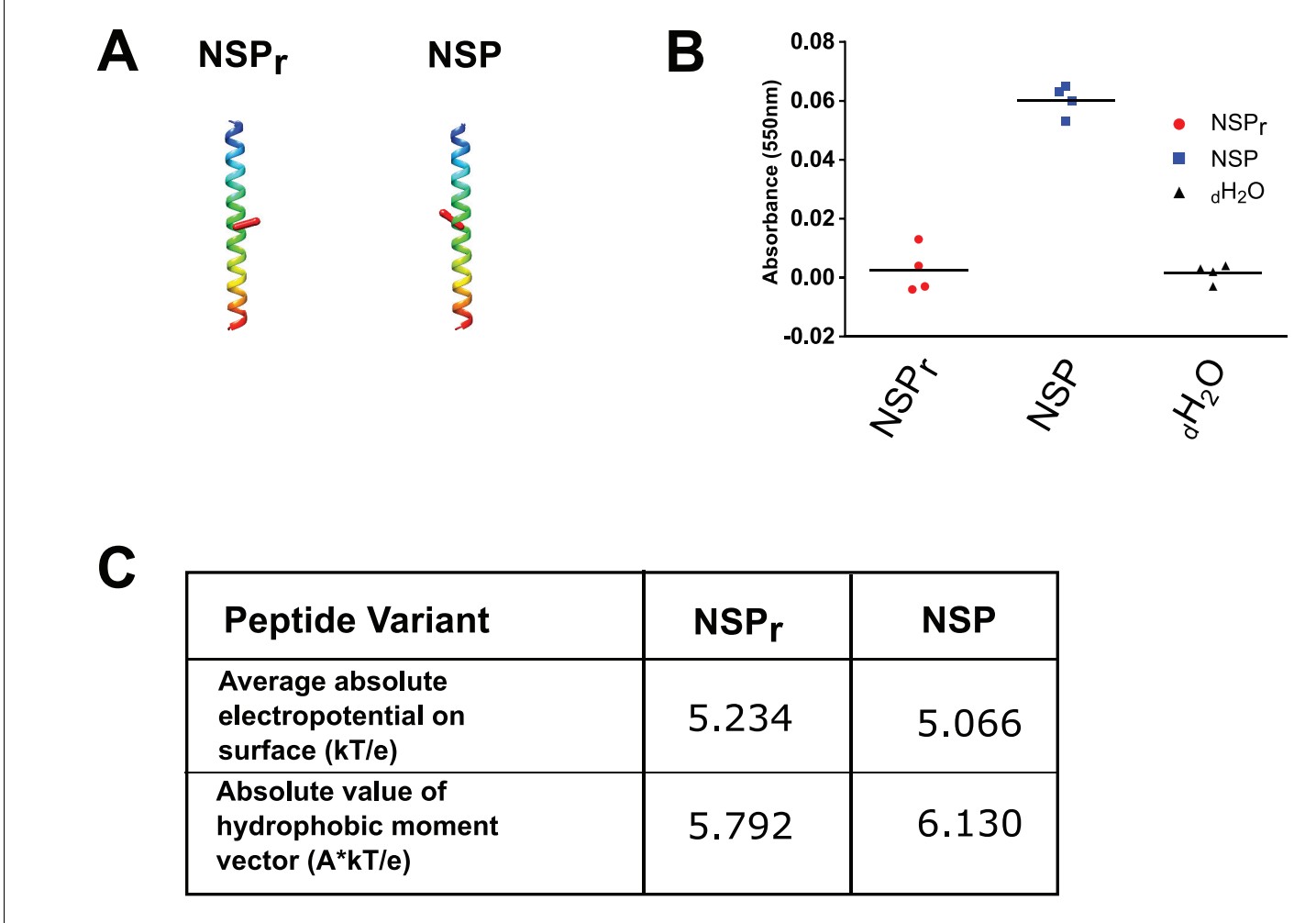

**Figure 1.** Solubility test of NSP and NSP$_r$. (**A**) Peptide models computed by the 3D-hydrophobic moment peptide calculator. The direction of hydrophobic moment is indicated by a red line. Peptides are oriented with their N to C-terminus from bottom (red) to top (blue). (**B**) Turbidity measurement of peptide suspension. The absorbance of light at 550 nm for NSP (blue squares, 15 mg/mL) and NSP$_r$ (red circles, 25 mg/mL), re-suspended in distilled water ($_d$H$_2$O) were compared to a $_d$H$_2$O control (green triangles). (**C**) Calculated electropotential and hydrophobic moment of peptide variants. Calculations were performed using the 3D-hydrophobic moment peptide calculator as described in Materials and methods.
DOI: https://doi.org/10.7554/eLife.34085.003

and colicin M is modulated by the ligand ferricrocin (**Figure 7B**), as previously reported in vivo and in vitro (**Mills et al., 2014**; **Wayne et al., 1976**). The peptidisc is therefore suitable for the functional reconstitution of both α-helical and β-barrel membrane proteins.

The 'in-gel' method (**Figure 8**) was developed to determine optimal reconstitution conditions in a time- and cost-effective manner. Small amounts of peptides (0–2.5 µg) were mixed with the target protein (~1.25 µg) in detergent solution, and the resulting mixture immediately loaded on native gel. In that case, removal of the non-ionic detergent occurs during electrophoresis when the protein-peptide mixture enters the detergent-free part of the gel. At the correct NSP ratio, the target membrane protein does not aggregate at the top of the gel but instead migrates in a soluble form to its expected molecular weight position. This simple method allowed us to estimate the effective NSP concentrations required to trap four different integral membrane complexes into a peptidisc: MalFGK$_2$ (**Figure 8A**), FhuA (**Figure 8B**), the trimeric OmpF porin (**Figure 8C**), and the membrane translocon SecYEG (**Figure 8D**). These membrane proteins were generally reconstituted at similar peptide concentrations, with a half-maximal molar ratio (RR50) of 20 (**Figure 8E and F**). This is significantly higher than the measured stoichiometry of ~10 peptides per protein complex (**Table 1**),

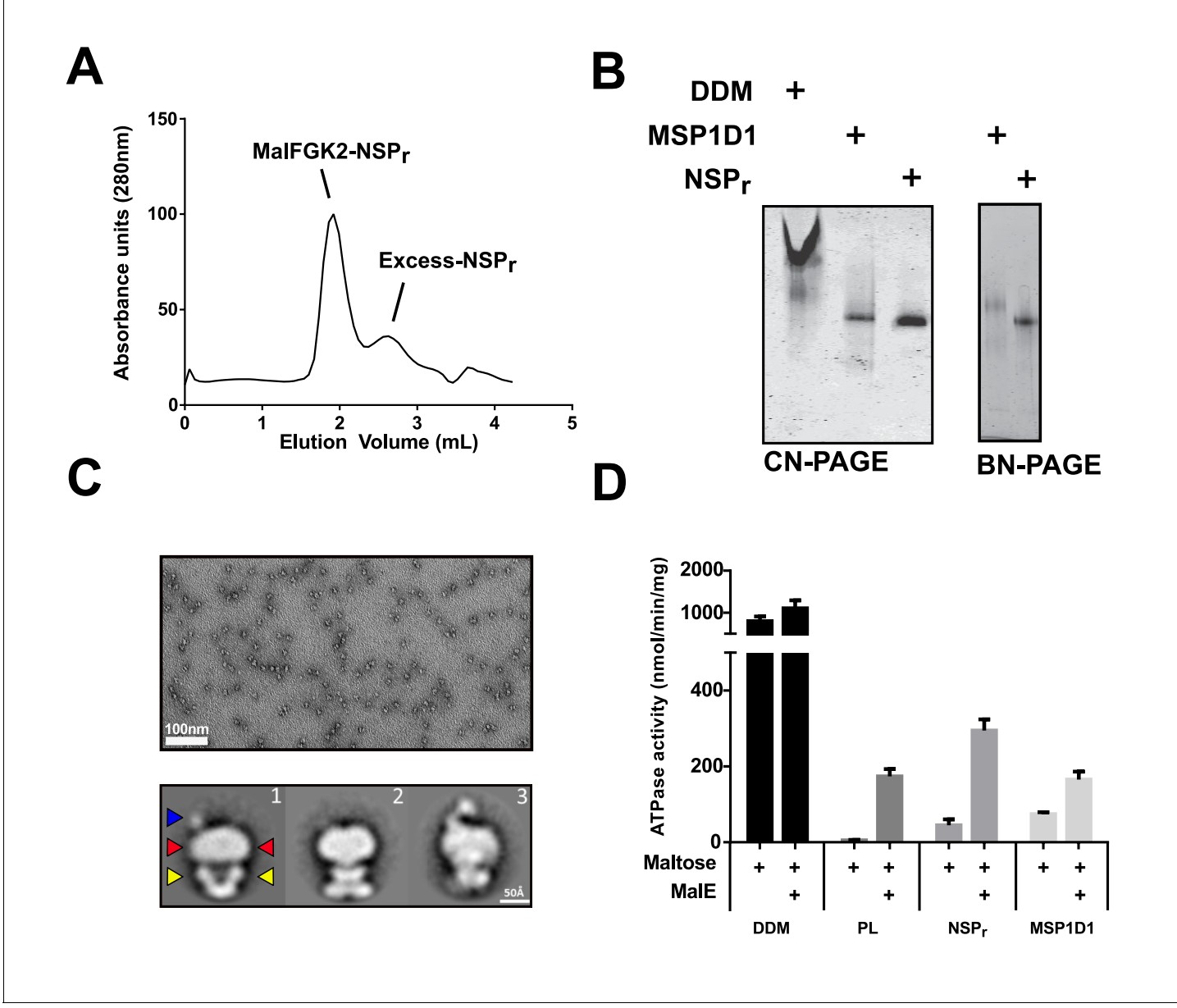

**Figure 2.** The 'on-column' reconstitution of MalFGK₂. (A) Typical size-exclusion chromatography of MalFGK₂ in peptidisc (MalFGK₂-NSP$_r$) using the 'on-column' method. (B) CN-PAGE and BN-PAGE analysis of MalFGK₂ in detergent micelle (DDM), nanodisc (MSP1D1), and peptidisc (NSP$_r$). (C) Top panel: Field of view of particles stained with uranyl formate. Bottom panel: Selected class averages representing three characteristic views of MalFGK₂ in peptidisc. The nucleotide-binding domains (MalK₂), the transmembrane domain (MalFG), and periplasmic P2-loop are indicated with yellow, red and blue arrows, respectively. (D) Maltose-dependent ATPase activity of MalFGK₂ (0.5 µM) reconstituted in detergent (DDM), proteoliposomes (PL), peptidiscs (NSP$_r$), and nanodiscs (MSP1D1) obtained at 30°C in the presence or absence of MalE (2.5 µM). Error bars represent standard deviations from three separate experiments.

DOI: https://doi.org/10.7554/eLife.34085.004

suggesting that excess peptide is needed to achieve efficient assembly. This analysis also showed that the SecYEG complex can be trapped as a dimer and higher order oligomeric form in peptidisc (*Figure 8D*), probably due to the self-association of this complex in detergent solution, as shown before (*Bessonneau et al., 2002*). This later observation further differentiates the peptidisc from the nanodisc. In the nanodisc, the selective reconstitution of the SecYEG monomer and dimer requires MSP proteins of different lengths (*Figure 8—figure supplement 1*) (*Dalal et al., 2012*).

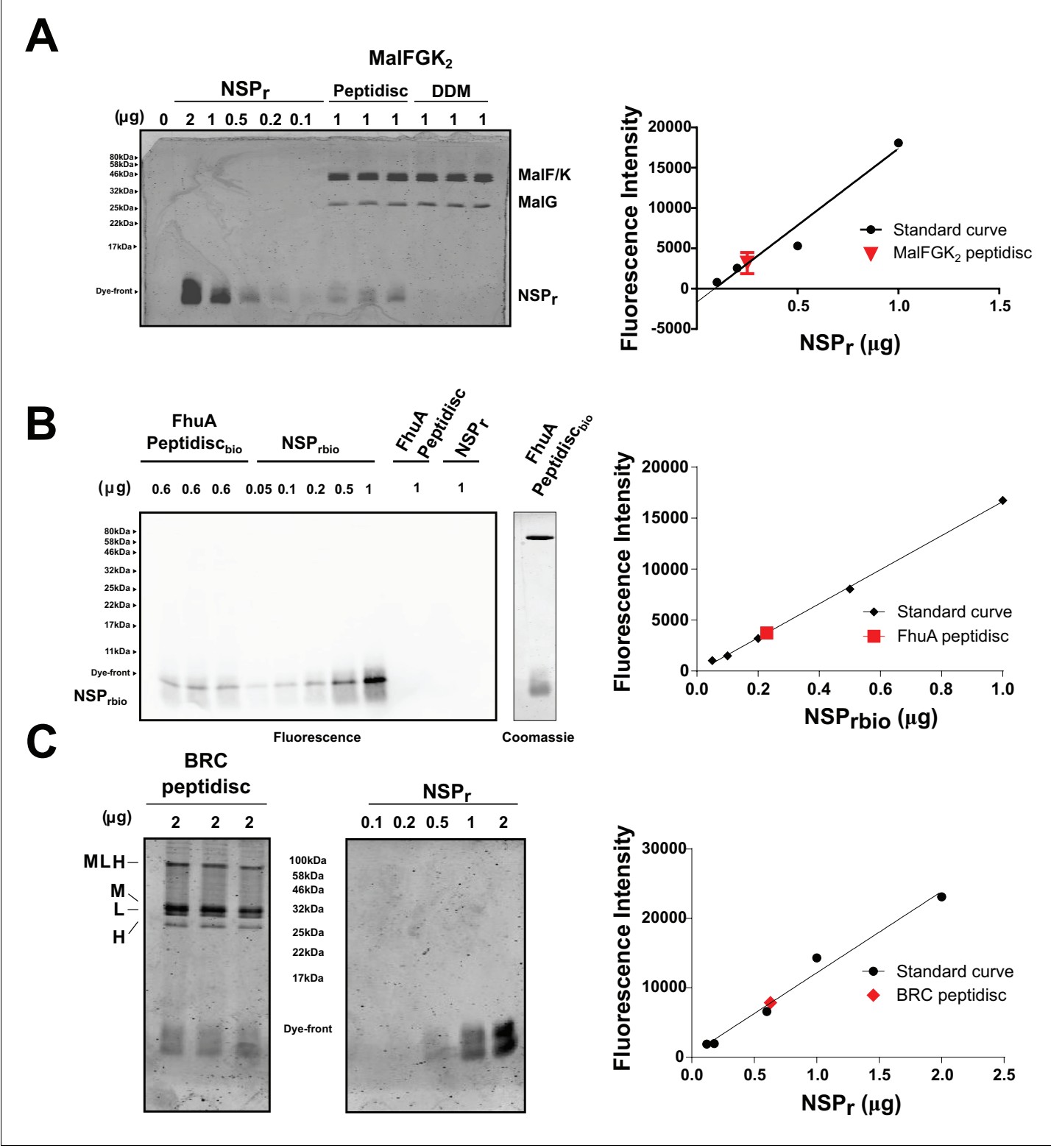

**Figure 3.** Quantification of NSP$_r$ in peptidiscs. (**A**) Left panel; 15% SDS-PAGE analysis of MalFGK$_2$ in peptidisc or DDM. NSP$_r$ runs at the bottom of the gel and can be visualized with Coomassie blue staining. Dye fluorescence was measured on a LICOR Odyssey scanner and quantified by Image J. Right panel; Standard curve derived from NSP$_r$ titration measurement (black dots), and average intensity of NSP$_r$ fluorescence from MalFGK$_2$ peptidisc (red dot). (**B**) Left Panel: Western Blot of FhuA-peptidisc reconstituted into NSP$_{rbio}$, and visualized by incubation with Streptavidin-Alexa 680. Fluorescence of the Alexa 680 dye was measured on a LICOR Odyssey scanner (700 nm, excitation 680 nm) and quantified in Image J. Right Panel: Standard curve as in

*Figure 3 continued on next page*

*Figure 3 continued*

A. (C) 15% SDS-PAGE analysis of BRC in peptidisc. The MLH subunits of BRC partially resist denaturation by SDS, resulting in a higher molecular weight band located above the single subunits. Each gel was repeated in triplicate with independent standard curves to calculate the values reported in *Table 1*.

DOI: https://doi.org/10.7554/eLife.34085.005

To save on peptide consumption, as well as to minimize exposure of the target to detergent, we developed the 'on-beads' reconstitution method (*Figure 9*), wherein affinity-purification of the protein and incorporation into peptidisc are carried out simultaneously. As illustrated in *Figure 9A*, the peptide was added in excess while the membrane protein still bound to the beads, followed by detergent dilution and eventually elution in a detergent-free buffer (*Figure 9A*, step 5). The method was tested using the his-tagged $MalFGK_2$ complex (*Figure 9B*). Analysis of the eluted complex by BN-PAGE and CN-PAGE showed that $MalFGK_2$ is readily incorporated into peptidiscs (*Figure 9C*), with purity and yield as good as with conventional detergent-based chromatography (*Figure 9C*). As an added advantage, the excess peptide collected before the elution step was re-used for the next round of 'on-beads' reconstitution, thereby reducing the cost of large-scale reconstitution.

Finally, we reconstituted the photosynthetic bacterial reaction center (BRC) from *Rhodobacter sphaeroides*, given its potential for biotechnological application. The BRC is employed in bio-hybrid solar cells due to its ability to absorb light in the near-infrared with high quantum efficiency (*Blankenship et al., 2011*; *Ravi and Tan, 2015*; *Yaghoubi et al., 2017*). However, sustained heat and light exposure lead to irreversible loss of pigments and protein

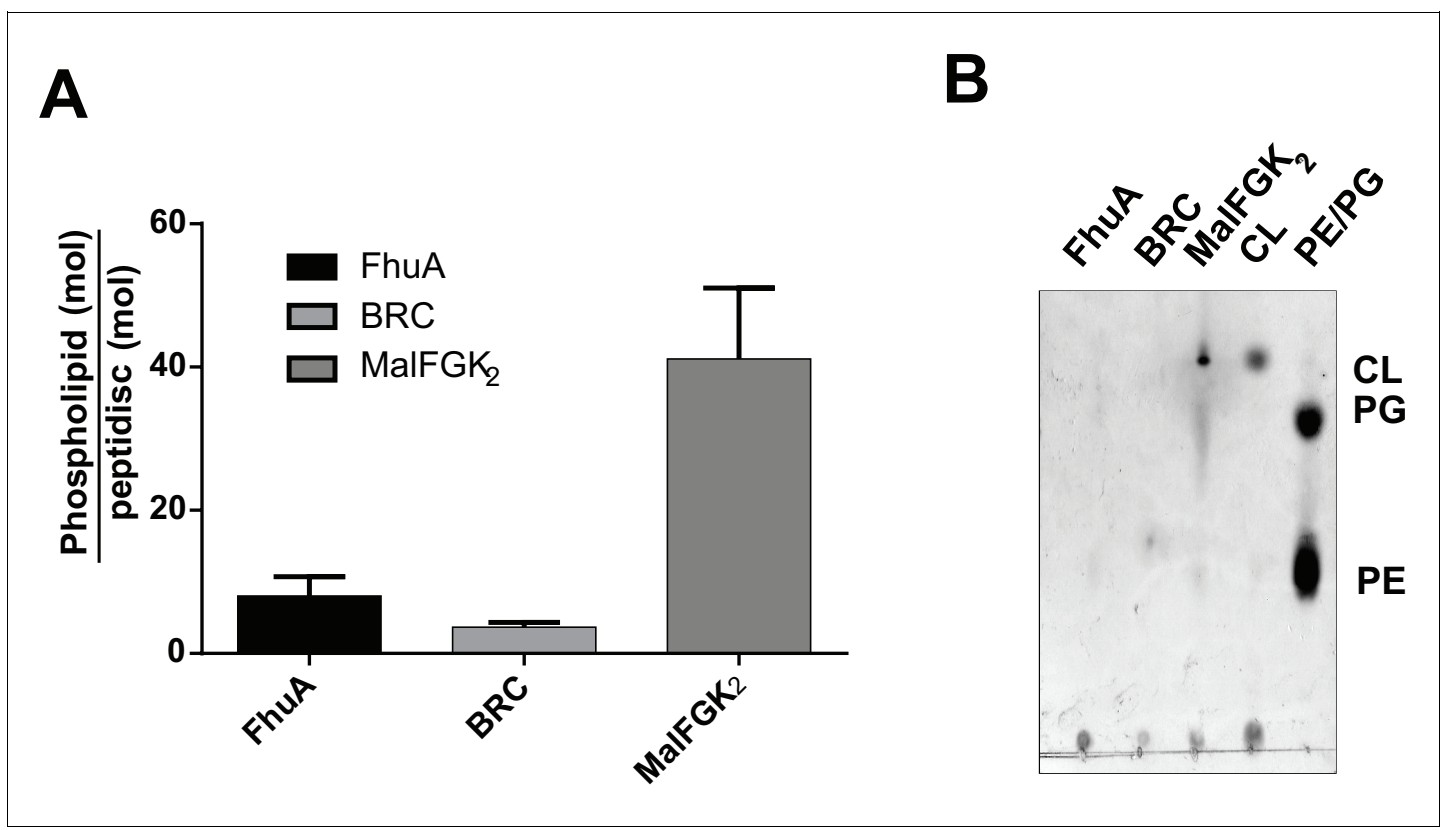

**Figure 4.** Quantification of phospholipids trapped in peptidiscs. (A) Calculated number of phospholipids per peptidisc. Phospholipid content was determined by Malachite green assay after acid digestion of lipid extracts. Error bars represent standard deviation derived from three separate measurements. B) TLC analysis of lipid extracts obtained from 10 µg $MalFGK_2$ peptidisc, 10 µg FhuA peptidiscs and 20 µg BRC peptidiscs, as well as pure lipid standards Cardiolipin (CL), 1,2-dioleoyl-sn-glycero-3-phosphoglycerol (PG), and 1,2-dioleoyl-sn-glycero-3-phosphoethanolamine (PE).

DOI: https://doi.org/10.7554/eLife.34085.006

**Table 1.** Calculated and observed molecular weight and scaffold stoichiometry of peptidiscs

| Peptidisc | Molecular weight measured by ESI-MS (kDa) | Measured NSP$_r$ stoichiometry (NSP$_r$/disc) | Measured lipid stoichiometry (Lipid/Disc) | Calculated molecular weight* (kDa) |
|---|---|---|---|---|
| MalFGK$_2$-NSP$_r$ | 247 ± 24 | 10 (±2): 1 | 41 (±10): 1 | 251 ± 12 |
| BRC-NSP$_r$ | 138 ± 18 | 9 (±1): 1 | 4 (±1): 1 | 138 ± 5 |
| FhuA-NSP$_r$ | 137 ± 18 | 10 (±2): 1 | 8 (±3): 1 | 131 ± 9 |

*The formula for the calculated molecular weight is as follows: MW$_{peptidisc}$ = MW$_{(protein)}$+n(MW$_{NSPr}$)+m(MW$_{Lipid}$); where n is the measured NSP$_r$ stoichiometry, m is the measured lipid stoichiometry, MW$_{Lipid}$ = 0.8 kDa, MW$_{NSPr}$ = 4.5 kDa, and MW$_{protein}$ = 173 kDa, 80 kDa, and 94 kDa for MalFGK$_2$, FhuA, and BRC, respectively. For NSP$_r$ and Lipid stoichiometry, the standard deviation is derived from three separate measurements.

DOI: https://doi.org/10.7554/eLife.34085.009

denaturation (*Hughes et al., 2006*; *Scheidelaar S et al., 2014*). Thermal stability and solubility at high protein concentration are therefore important parameters for successful application. The molecular weight of the BRC peptidisc measured by native mass spectrometry is 138 ± 18 kDa (*Table 1*, *Figure 5C and D*). It has been shown that purification of BRC in LDAO delipidates the complex (*Scheidelaar S et al., 2014*), and accordingly the BRC peptidisc contained only 4 ± 1 phospholipids (*Table 1*, *Figure 4A*). Analysis by SDS-PAGE indicated a stoichiometry of 9 ± 1 NSP per BRC (*Table 1*, *Figure 3C*). The calculated molecular weight was 138 ± 5 kDa, in excellent agreement with native mass spectrometry data (*Table 1*). We next measured the stability of the BRC pigments in peptidiscs or in LDAO detergent (*Figure 10—figure supplement 1A and B*). The spectral properties of the BRC complex were similar in both environments (*Figure 10A*). However, the BRC in peptidisc resisted denaturation 65°C for 1 hr, while it was fully denatured in less than 4 min in LDAO (*Figure 10B*). This difference corresponds to ~100 fold increase of the half-life of the BRC complex at elevated temperatures (*Figure 10C*).

We also compared the thermal stability of the BRC complex when reconstituted without additional lipids in SMA polymer and nanodiscs (*Figure 10—figure supplement 1C and D*). Without added lipid, the diameter of the BRC is too small for the MSP1D1 belt, resulting in some protein aggregation and heterogenous nanodisc preparation (*Figure 10—figure supplement 1F*, left panel, lane 4). Reconstitution into the SMA polymer without lipids was monodisperse (*Figure 10—figure supplement 1F*, left panel, lane 5) and the thermostability was higher than in LDAO (*Figure 10—figure supplement 1E*). This observation is contrary to previous reports which suggest that the lipid environment in the SMA particle is what increases BRC thermostability. (*Scheidelaar S et al., 2014*) In our hands, reconstitution of the BRC into peptidiscs, proteoliposomes, low-lipid nanodiscs and SMA particles all result in comparable thermostability (*Figure 10—figure supplement 1E*). From these results, it appears more important to remove detergents than include lipids to increase thermal tolerance of the BRC.

## Discussion

Detergents remain the most effective way to extract and purify membrane proteins, yet these surfactants have many undesired effects on protein stability and downstream biochemical analysis. To circumvent these difficulties, and to handle these proteins like their water-soluble counterparts, membrane proteins are more often reconstituted with amphipathic scaffolds. However, current methods of reconstitution are difficult because each scaffold system has specific properties and limitations, and each requires substantial optimization. The aim of our work was to develop a 'one-size fits all' method to streamline the capture of membrane proteins in detergent-free solution.

We show that the peptidisc is a simple and efficient way to replace detergent. The system works with membrane proteins of different size, fold and complexity, and does not require addition of exogenous lipids. The peptidisc captures membrane proteins regardless of their initial lipid content, and therefore the use of exogenous lipids to match the diameter of the scaffold such as in the nanodisc system is avoided. Since the binding of the peptide is essentially guided by the size and shape of the protein template, the peptide stoichiometry is also self-determined. As a direct consequence, the preparation of peptidisc is possible through rapid detergent removal techniques such as 'in-gel',

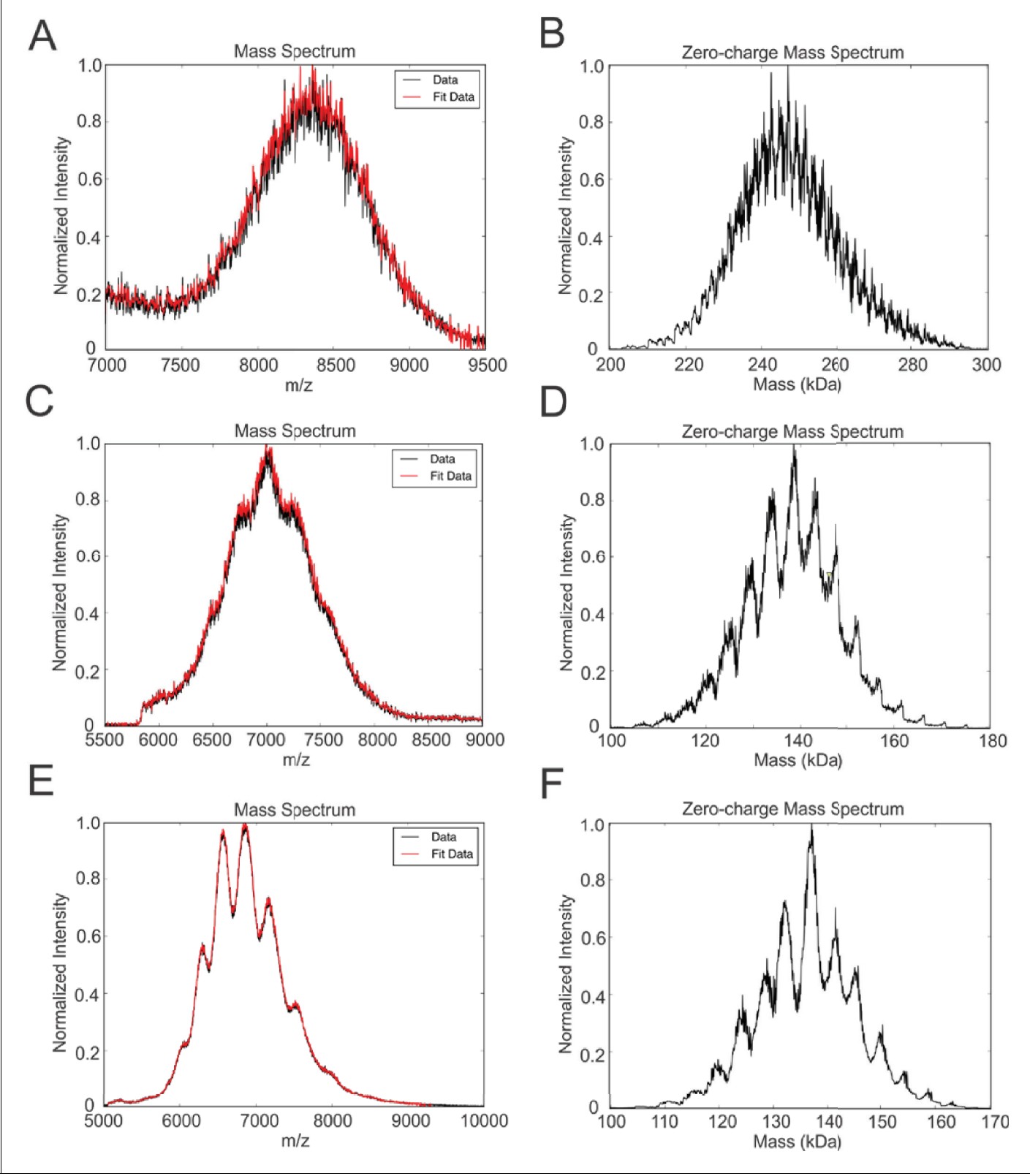

**Figure 5.** Native mass spectrometry of intact peptidiscs. Panels A, C, E are mass spectra acquired in positive ion mode for aqueous ammonium acetate solutions (100 mM, pH 7, 22°C) of MalFGK$_2$-NSP$_r$, BRC-NSP$_r$ and FhuA-NSP$_r$, respectively. Panels B, D, and F are deconvoluted mass spectra of the peptidiscs shown in A, C, and E, respectively.

DOI: https://doi.org/10.7554/eLife.34085.007

**Table 2.** Molecular weight, diameter, and scaffold stoichiometry of MalFGK$_2$ reconstituted in peptidisc.

| | Measured molecular weight (kDa)* | Measured diameter $^\dagger$ | Calculated stoichiometry (scaffold/disc)$^\ddagger$ |
|---|---|---|---|
| MalFGK$_2$ Peptidisc | 250 ± 17 | 11.7 ± 1.4 | 12 (±2):1 |

*Molecular weight calculated from SEC-MALS data (Fig. S1). The standard error is derived from three independent SEC-MALS experiments. †Diameter of MalFGK$_2$-peptidiscs determined by negative stain electron microscopy **Figure 1B**, assuming a perfectly circular shape. ‡Stoichiometry (n), based on the measured diameter of the particles, was calculated with the following formula: $\pi(d_{disc}-2d_{\alpha\text{-helix}}) = (n/2)L_{NSPr}$; where $d_{\alpha\text{-helix}}$ represents the diameter of an alpha-helix (0.5 nm), $d_{disc}$ represents the measured disc diameter, and $L_{NSPr}$ represents length of the NSP$_r$ peptide.

DOI: https://doi.org/10.7554/eLife.34085.010

'on-column' and 'on-bead' methods. Each of these methods has considerable advantages compared to overnight dialysis and 'biobeads' techniques traditionally used for scaffold reconstitution. The 'in-gel' method is fast, high throughput and requires only 1–2 µg of the precious target protein in order to identify the optimal peptide ratio (*Figure 8*). The 'on-column' method allows direct preparation of large quantities of peptidiscs with simultaneous removal of salts, detergent and excess peptides. However, gel filtration requires concentrated protein samples and unbound peptide is wasted. The 'on-bead' method is therefore ideal for the peptidisc reconstitution because protein purification, detergent removal, trapping in peptidisc and recovering of unbound peptide are carried out simultaneously in the same tube. Additionally, because long exposure or storage of the protein in detergent is avoided, the 'on-bead' method can be especially advantageous in the case of unstable membrane proteins.

Beside these methodological considerations, we show that the peptidisc maintain proteins in a functional state. For instance, both FhuA and MalFGK$_2$ retain their ability to interact with their soluble binding partners. In contrast to the detergent, the ATPase activity of MalFGK$_2$ in peptidisc regains dependence to substrate and maltose-binding protein MalE, indicating a return to the

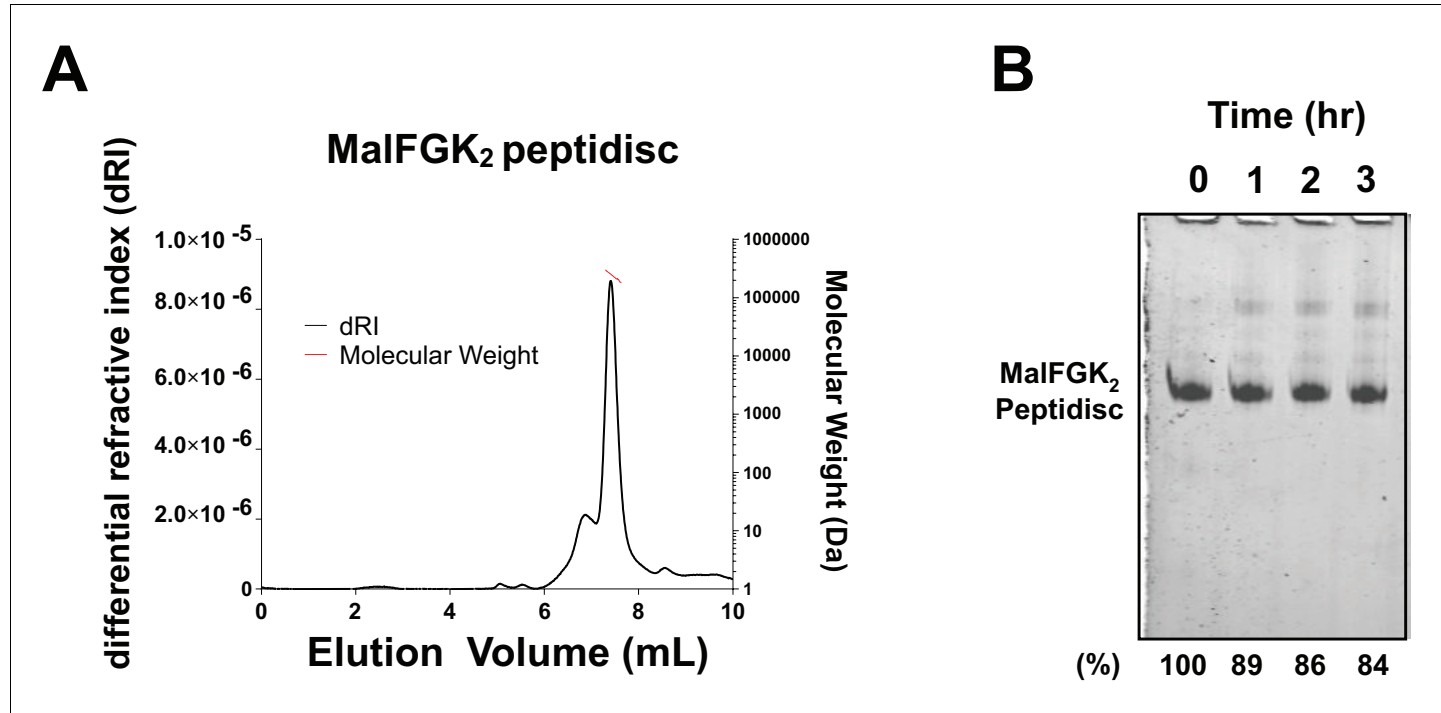

**Figure 6.** Stability of MalFGK$_2$ peptidisc. (**A**) Multi-angle light scattering analysis of MalFGK$_2$ reconstituted in peptidisc. MalFGK$_2$-NSP$_r$ (100 µg) was left for 3 days at 4°C before analysis by SEC-MALS. Protein sample was injected and protein concentration tracked through differential refractive interferometry (dRI, black trace). Molecular weight was calculated for the fractions corresponding to the peak of MalFGK$_2$-NSP (red trace). (**B**) Structural stability of MalFGK$_2$-NSP$_r$. The MalFGK$_2$ peptidisc was incubated at 30°C in Buffer A for the indicated time, then analyzed by BN-PAGE.

DOI: https://doi.org/10.7554/eLife.34085.008

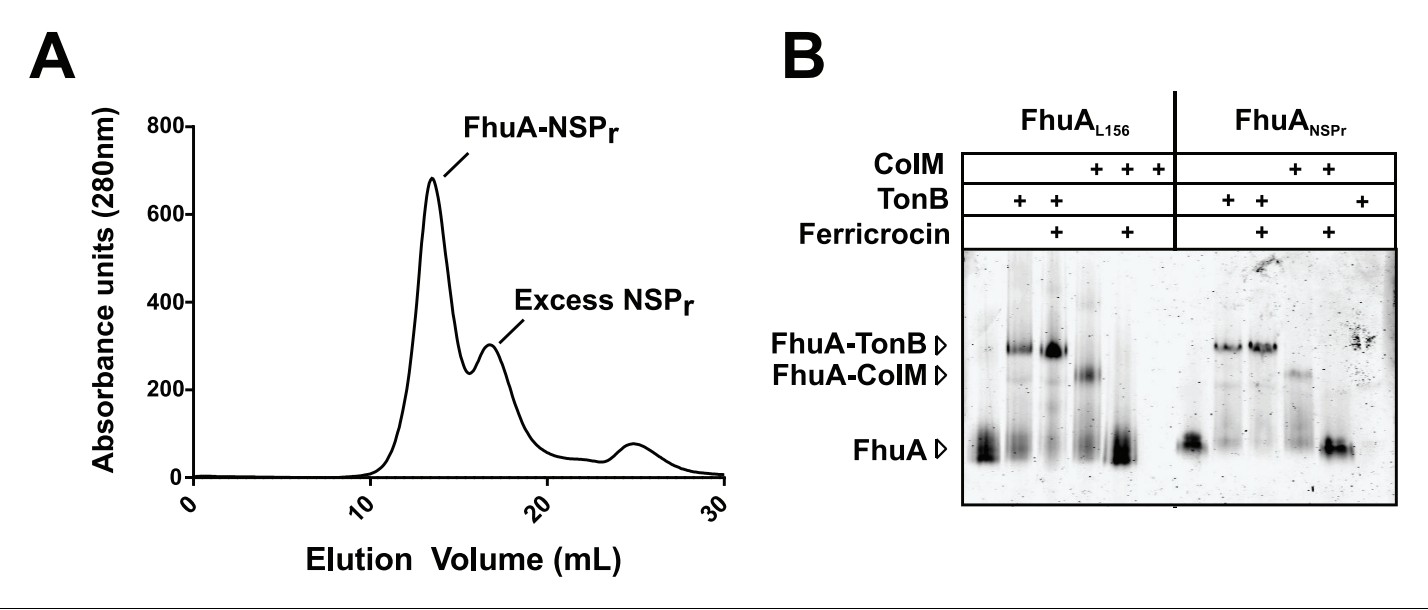

**Figure 7.** Binding activity of FhuA in nanodiscs and peptidiscs. (A) Typical SEC profile of FhuA reconstituted in peptidisc (FhuA-NSP$_r$) using an 'on-column' reconstitution protocol as described in Materials and methods. (B) The FhuA transporter reconstituted in nanodiscs (FhuA-MSP$_{L156}$) or peptidiscs (FhuA-NSP$_r$) was incubated with the C-terminal TonB$_{23-329}$ fragment (2 µg) or with colicin M (5 µg), with or without ferricrocin as indicated. Samples were analyzed by CN-PAGE and Coomassie-blue staining of the gel.

DOI: https://doi.org/10.7554/eLife.34085.011

transporter's native membrane conformation (*Bao and Duong, 2012*). Negative-stain electron microscopy of MalFGK$_2$ shows good resolution for the periplasmic P2 loop and cytosolic ABC domains, whereas the transmembrane domains MalFG are surrounded by an extra density corresponding to the peptide belt and naturally present lipids (*Figure 2C*). Lipid quantitation indeed indicates that ~ 40 lipids molecules, mostly acidic, are captured with MalFGK$_2$ (*Table 1*, *Figure 4A*). In the case of the BRC complex however, there must be direct protein-peptide contact because almost no lipids are detected in the final assembly (*Table 1*, *Figure 4A and B*). Despite the absence of lipids, the thermal stability of the BRC in peptidisc is still much higher than in detergent, with a melting temperature similar to that observed proteoliposomes (*Figure 10—figure supplement 1E*). Clearly, the peptidisc is more than a simple surrogate of the detergent molecules; the peptide assembly forms an environment that stabilize and support the transmembrane domains of a membrane protein from aqueous solution.

The peptidisc offers distinct advantages when compared to other amphipathic scaffolds. Unlike beltides, peptergents, lipopeptides, and nanostructured [beta]-sheet peptides, the NSP$_r$ does not require modifications at its N- or C-termini (*Corin et al., 2011*; *Tao et al., 2013*; *Privé, 2009*; *Larsen et al., 2016*). Both ends of NSP$_r$ remain available for modifications, such as biotinylation, which we show does not affect peptidisc assembly (*Figure 3B*). Importantly, peptergents and lipopeptides have critical micelle concentrations and dynamically exchange in solution. Because of this instability, an excess of these peptides is always needed to keep membrane proteins in solution (*Corin et al., 2011*; *McGregor et al., 2003*). This is not the case for the peptidisc because NSP$_r$ binds strongly to its target, allowing free peptides to be removed without compromising peptidisc stability. The length of the NSP$_r$ also does not need to be adjusted to the diameter of the target protein. This property may be especially advantageous when capturing macromolecular membrane protein complexes or proteins that exist in oligomeric state, as is the case with the SecYEG complex (*Figure 8—figure supplement 1B*). Lastly, the NSP$_r$ is synthetically made and can be obtained in large quantity, high purity and free of immunogens usually found in recombinant cell expression systems (*Schwarz et al., 2014*). The structural homogeneity of NSP$_r$ is also high compared to other synthetic scaffolds, such as amphipols and styrene-maleic acids (SMA). This is because peptide synthesis is sequential, whereas addition of carboxylate and other repeating units in amphipols and

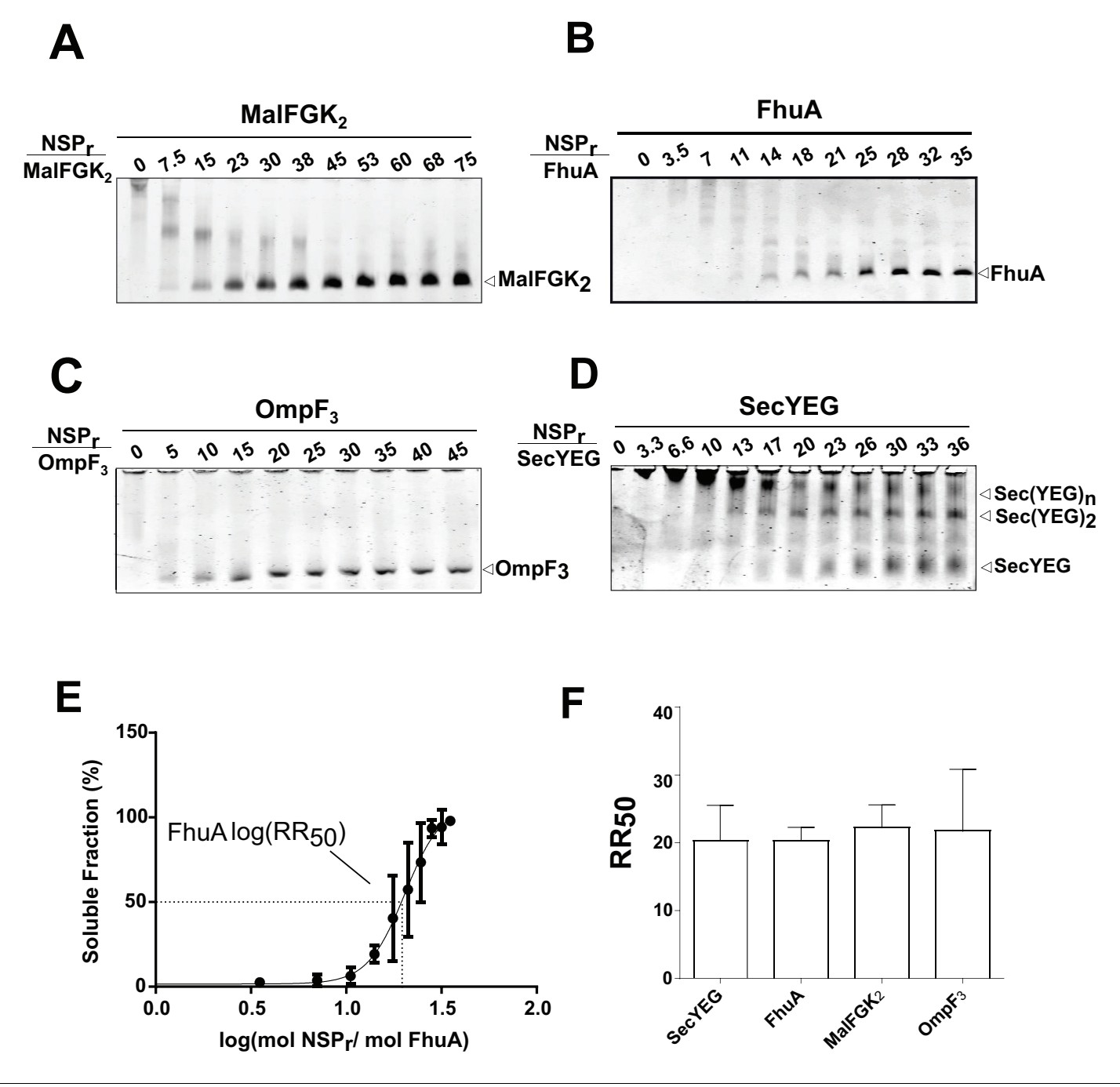

**Figure 8.** Express 'in-gel' method for determining optimal reconstitution ratio. (**A**) NSP and MalFGK$_2$ were mixed at the indicated molar ratio for 2 min in Buffer A containing a low amount of detergent (~0.008% DDM) before loading onto CN-PAGE. Peptidisc reconstitution occurs during migration in the detergent-free gel environment. The same experiment was performed with. (**B**) FhuA,. (**C**) OmpF$_3$,. (**D**) SecYEG. (**E**) Reconstitution efficiency of FhuA as a function of the NSP concentration. The protein band FhuA-NSP in B) was quantified with Image J and the data plotted as log (mol NSP/mol FhuA). The data were fitted with a Boltzmann sigmoidal function to generate a curve describing the reconstitution efficiency and the half-maximal reconstitution ratio (RR50). (**F**) The RR50 was determined for other target proteins as in (**E**). Error bars represent the standard deviation from three separate reconstitution experiments.

DOI: https://doi.org/10.7554/eLife.34085.012

The following figure supplement is available for figure 8:

**Figure supplement 1.** Capture of SecYEG monomer and dimer in peptidisc and nanodisc.

DOI: https://doi.org/10.7554/eLife.34085.013

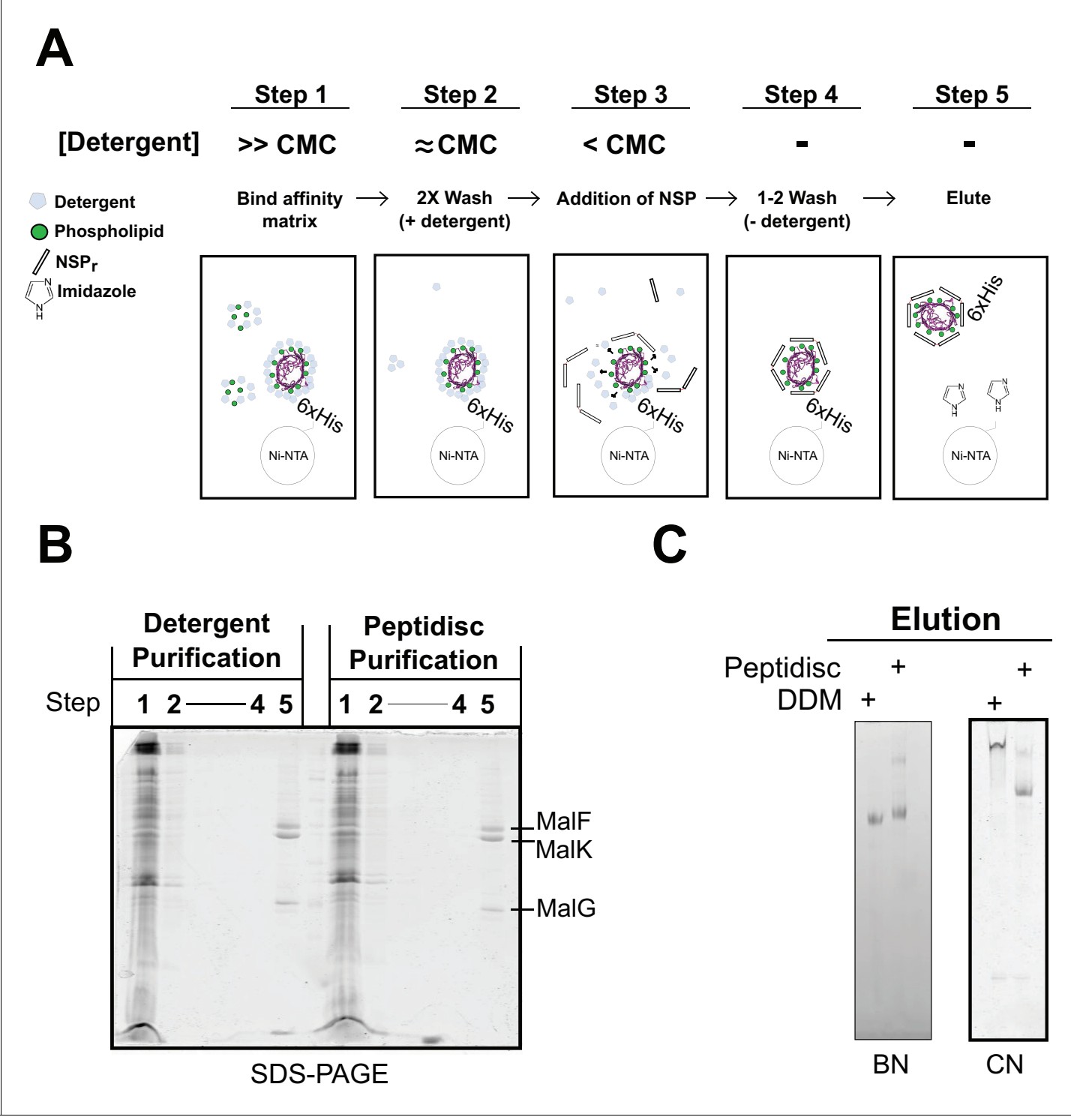

**Figure 9.** Direct 'on-beads' reconstitution during membrane protein purification. (**A**) Principle of the 'on-beads' reconstitution. Step 1: the tagged protein is extracted from the membrane with excess of detergent buffer (>CMC) and incubated with the affinity resin. Step 2: The beads are washed twice with the detergent buffer near its critical micelle concentration (~CMC). Step 3: The beads are incubated with buffer containing excess NSP and limited amount of detergent (<CMC). Step 4: The beads are washed in detergent-free buffer to remove unbound NSP and residual detergent. Step 5: The protein captured in peptidiscs is eluted from the column in detergent-free solution. (**B**) SDS-PAGE and. (**C**) Native-PAGE analysis of the his-tagged MalGFK$_2$ complex purified following conventional detergent method and 'on-beads' peptidisc detergent-free method.

DOI: https://doi.org/10.7554/eLife.34085.014

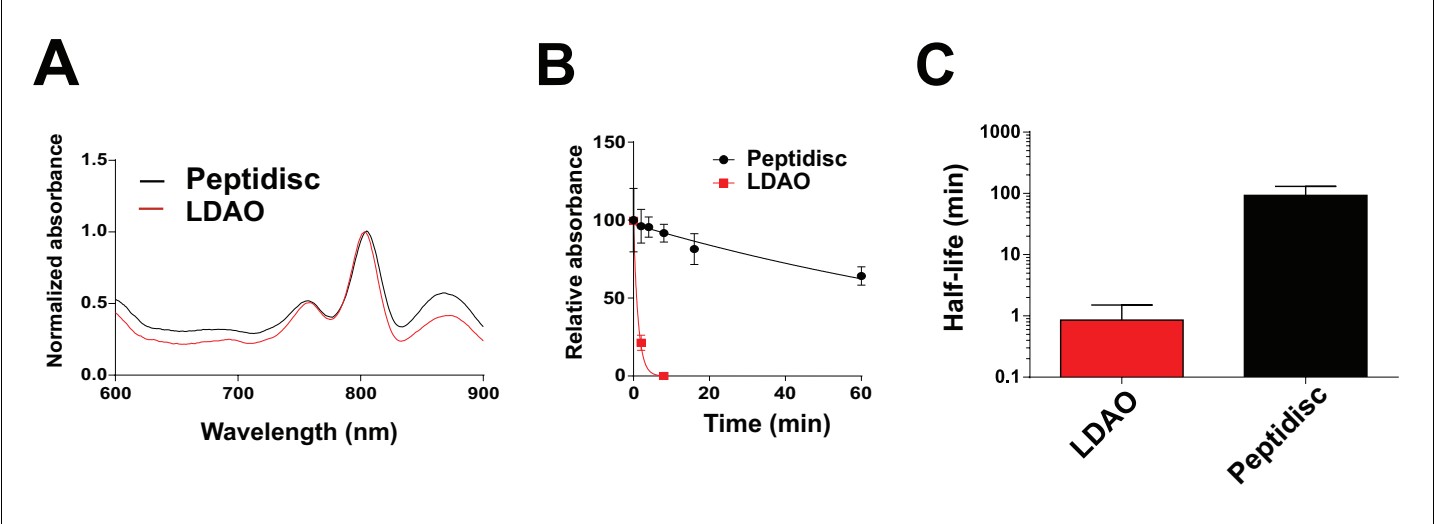

**Figure 10.** Thermostability of the BRC complex in peptidiscs. (**A**) Absorbance scans of the BRC (1 μM) in detergent solution (0.03% LDAO, red trace) and in peptidisc (black trace). Scans were normalized to the value measured at 803 nm (the absorbance peak of the accessory bacteriochlorophylls). (**B**) Decrease in absorbance of the BRC at 803 nm after incubation at 65°C for the indicated time. (**C**) Calculated half-life of the BRC in peptidisc and LDAO at 65°C. The data in B) were fit with an exponential decay function to determine the corresponding half-life. Error bars represent the standard deviation from three separate experiments.

DOI: https://doi.org/10.7554/eLife.34085.015

The following figure supplement is available for figure 10:

**Figure supplement 1.** Effect of peptidisc on BRC stability.
DOI: https://doi.org/10.7554/eLife.34085.016

industrially prepared SMA polymers is randomly distributed along the polymer chain during assembly. Due to the 'one-pot' synthesis, these synthetic polymer preparations are often polydisperse mixtures of different lengths (*Popot, 2010*; *Smith et al., 2017*). This compositional heterogeneity can be problematic for functional and structural studies (*Deller et al., 2016*). Finally, due to both positive and negative charged amino acids, the solubility of the NSP$_r$ remains high at various pH or with divalent cations. Both these conditions can destabilize assemblies formed by synthetic polymers because they largely depend on charged carboxylate groups for solubility (*Popot, 2010*; *Gulati et al., 2014*).

How exactly the peptidisc wraps around the membrane protein template remains an important question. In ApoA1 lipid nanodiscs, the two scaffold proteins arrange themselves in an anti-parallel 'double belt' configuration (*Bibow et al., 2017*). If NSP$_r$ were also arranged in a double belt, then the peptide to protein ratio would expectedly vary by factor of 2, as an odd number of scaffold would leave part of the protein exposed to the environment. However, the native mass spectrometry profiles for FhuA and BRC indicate peptidisc populations which can differ in mass by one peptide only (*Figure 5D and F*), suggesting an arrangement that is flexible. Possibly, the NSP$_r$ could be arranged in an orthogonal 'picket fence' orientation as proposed for lipopeptides, nanostructured [beta]-sheet peptides, and single helix ApoA1 mimetic peptides.(*Tao et al., 2013*; *Privé, 2009*; *Islam et al., 2018*). However, the length of NSP$_r$ (37 amino acids) is too long to be orthogonal while maintaining contact with hydrophobic parts of the protein or alkyl chains of annular lipids. Thus, the NSP$_r$ perhaps simply lies in a tilted orientation. A tilted orientation would facilitate optimal binding and stoichiometry as the peptide shifts in angle of association, adapting to best fit the target membrane protein template.

In conclusion, the peptidisc offers several advantages. The method is cheap, fast and seamlessly integrated in existing protein purification protocols, such as size exclusion and affinity chromatography. The peptide is relatively simple to synthesize and it can be recycled via the 'on-bead' method to decrease consumption. The peptidisc is not hindered by issue of buffer instability or heterogeneity as observed with other synthetic scaffold. Since the peptide self-associates to its template and without added lipids, this could be advantageous for structural studies which are affected by compositional heterogeneity. There are of course applications where other scaffold systems are better

suited, such as direct protein solubilization with SMA polymers, or control over lipid environment offered by nanodiscs. Nevertheless, the current advantages of the peptidisc surely diminish the challenges associated with biochemical, structural and pharmacological characterization of purified membrane proteins. The peptidisc may emerge as a very practical way to analyse or exploit membrane proteins in a detergent-free environment.

## Materials and methods

### Biological reagents

Tryptone, yeast extract, NaCl, imidazole, Tris-base, acrylamide 40%, bis-acrylamide 2% and TEMED were obtained from Bioshop, Canada. Isopropyl β-D-1-thiogalactopyranoside (IPTG), ampicillin, and arabinose were purchased from GoldBio. Detergents n-dodecyl-β-D-maltoside (DDM) and octyl-β-D-glucoside (β-OG) were from Anatrace. Detergent N,N-dimethyldodecylamine N-oxide (LDAO) was from Sigma. Total $E.coli$ lipids were purchased from Avanti Polar Lipids. Resource 15Q, Fast Flow S, Superdex 200 HR 10/300 GL and 5/150 GL were obtained from GE Healthcare. $Ni^{2+}$-NTA chelating Sepharose was obtained from Qiagen. All other chemicals were obtained from Fischer Scientific Canada.

### Peptides

Peptide $NSP_r$ ($N_{ter}$-FAEKFKEAVKDYFAKFWDPAAEKLKEAVKDYFAKLWD-$C_{ter}$) and NSP ($N_{ter}$- D WLKAFYDKVAEKLKEAAPDWFKAFYDKVAEKFKEAF-$C_{ter}$) were obtained from $A^+$ peptide Co. Ltd. and Genscript (each with purity >80%). Peptide $NSP_{rbio}$ was obtained from KareBay. The effective purity of the peptides employed is actually higher as mass spectrometry analysis finds that main contaminants (>10%) consists of peptides missing the final aspartate residue. To aid accessibility to the academic community, bulk $NSP_r$ peptides and core protocols are available at www.peptidisc.com

### Preparation of the NSP peptides

For solubility test experiments, lyophilized NSP and $NSP_r$ (purity of 82% and 85%, respectively) were resuspended in $_dH_20$ at room temperature to final concentrations of 15 mg/mL and 25 mg/mL, respectively. Peptide concentration was determined by absorbance at 280 nm. Residual TFA from peptide synthesis results in a low pH solution (pH 2–3). For all other experiments, peptides were solubilized in $_dH_2O$ at 6 mg/mL. Solubilized peptides were stored at 4°C for up to 5 weeks. Immediately before use, the pH of the peptide solution was modified by addition of 20 mM Tris-HCl, pH 8 to form the so-called Assembly Buffer. Immediately before use in peptidisc reconstitutions, peptide concentration in Assembly Buffer was verified by Bradford assay (*Prehna et al., 2012*).

### Protein expression and purification

Unless otherwise stated, all proteins were expressed in $E.coli$ BL21(DE3) (New England Biolabs) for 3 hr at 37°C after induction at an OD of 0.4–0.7 in LB medium supplemented with required antibiotic. Cells were harvested by low-speed centrifugation (10,000 x g, 6 min) and resuspended in Buffer A (50 mM Tris-HCl: pH 8; 100 mM NaCl; 10% glycerol). Resuspended cells were treated with 1 mM phenylmethylsulfonyl fluoride (PMSF) and lysed using a microfluidizer (Microfluidics) at 10,000 psi. Unbroken cell debris and other aggregates were removed by an additional low-speed centrifugation. Cytosolic and crude membrane fractions containing the overexpressed protein of interest were subsequently isolated by ultracentrifugation (100,000 x g, 45 min) and crude membrane fraction resuspended in Buffer A (50 mM Tris-HCl: pH8, 100 mM NaCl, 10% glycerol). MalE and His-tagged $MalFGK_2$ were purified as previously described (*Bao and Duong, 2012*), expressed from plasmids pBAD33-MalE and pBAD22-FGK$_{his}$, respectively. Crude membrane containing His-tagged $MalFGK_2$ were solubilized at 4°C overnight in Buffer A + 1% DDM and clarified by ultracentrifugation. Solubilized $MalFGK_2$ was isolated by $Ni^{2+}$-chelating chromatography in Buffer A + 0.02% DDM, washed in five column volumes (CV) of Buffer B (50 mM Tris-HCl: pH 8; 200 mM NaCl; 15 mM imidazole; 10% glycerol)+0.02% DDM, and then eluted in Buffer C (50 mM Tris-HCl: pH 8; 100 mM NaCl; 400 mM imidazole; 10% glycerol)+0.02% DDM. Protein MalE was isolated on Resource 15Q column, concentrated using a 30 kDa polysulfone filter (Pall Corporation), and then further purified on Superdex 200 HR 10/300 GL column equilibrated in Buffer EQ (50mM Tris-HCl: pH 8, 50 mM NaCl, 10% glycerol).

His-tagged-MSP$_{L156}$ and His-tagged TonB$_{23-329}$ were purified by Ni$^{2+}$-chelating chromatography as previously described (*Mills et al., 2014*). His-tagged Colicin M was expressed and purified according to established protocols from plasmid pMLD189 in the *E. coli* strain BW25113 (*Mills et al., 2014*). His-tagged FhuA, encoded by plasmid pHX405, was expressed in *E. coli* strain AW740 (Δ*ompF*, Δ*ompC*) in M9 minimal media and was purified in LDAO as previously described (*Mills et al., 2014*) OmpF was expressed from *E.coli* JW2203 (Δ*OmpC*) as previously described (*Jun et al., 2014*). Pre-pared crude membrane was resuspended in Buffer A and the inner membrane solubilized by addition of 1% Triton X-100. The outer membrane fraction (OM) was isolated by ultracentrifugation, resuspended in Buffer A + 1% LDAO at a concentration of 3 mg/mL, and incubated overnight at 4°C with gentle rocking. Insoluble material was removed by an additional ultracentrifugation step, and the clarified lysate was applied onto a Resource 15Q column pre-equilibriated in Buffer EQ + 0.1% LDAO. OmpF was eluted by a linear 20 mL gradient of 50–700 mM NaCl, and further purified by Superdex 200 HR 10/300 in Buffer A + 0.1% LDAO. Expression and purification of His-tagged SecYEG was performed from the plasmid pBad22-His-EYG as previously described (*Dalal et al., 2012*). Crude membranes were solubilized for 1 hr at 4°C in Buffer A + 1% DDM. Solubilized material was clarified by ultra-centrifugation and passed over a 5 mL Ni$^{2+}$-NTA column. After extensive washing in Buffer A + 0.02% DDM, SecYEG was eluted in Buffer A + 0.02% DDM over a 20 mL gradient of 0–600 mM imidazole. The most concentrated fractions were pooled and diluted fivefold in Buffer O (50 mM Tris-HCl: pH 8, 10% glycerol +0.02% DDM) before being applied to a 5 mL Fast Flow S cation exchange column pre-equilibrated in Buffer EQ + 0.02% DDM. Bound protein was eluted over a 20 mL gradient from 50 to 600 mM NaCl in Buffer EQ + 0.02% DDM. Plasmids pET28 encoding his-tagged MSPD1 and MSP1D1E3 proteins were transformed into BL21 cells and protein expression and purification was performed as previously described (*Dalal et al., 2012*). All proteins, with the exception of BRC, were flash frozen in liquid nitrogen immediately after purification and stored at −80°C for later use. BRC was purified as previously described (*Bradford, 1976*). In brief, His-tagged BRC was expressed in *Rhodobacter Sphaeroides* RcX (Δ*puhA*, Δ*pufQBALMX*, Δ*rshI*, Δ*ppsR*) using plasmid pIND4-RC1. A preculture of 10 mL in RLB media (LB medium; 810 μM MgCl$_2$; 510 μM CaCl$_2$)+25 μg/mL kanamycin was transferred into 100 ml of RLB-kan and grown overnight at 30°C before transfer into 1 L of freshly prepared RLB-kan. After growth for 8 hr at 30°C, BRC production was induced with 1 mM IPTG for an additional 16 hr. During growth and purification, light exposure was kept to a minimum. Cells were harvested by low-speed centrifugation, resuspended in Buffer A and lysed by French press (10,000 psi). Unbroken cells and cell debris were removed by low speed centrifugation, and the supernatant treated with 1% LDAO overnight at 4°C. After removal of insoluble material by ultracentrifugation, the supernatant was supplemented with 10 mM imidazole and the BRC purified by Ni$^{2+}$-chelating affinity chromatography. BRC bound to affinity resin was washed overnight at 4°C in 20 column volumes of Buffer B + 0.03% LDAO, before elution in Buffer C + 0.03% LDAO. The complex was further purified on a Superdex 200 HR 10/300 GL in Buffer A + 0.03% LDAO, and stored in the dark at 4°C before use in thermostability assays.

## 'On-column' peptidisc reconstitution

MalFGK$_2$ (300 μg) in Buffer E + 0.02% DDM was mixed with NSP (480 μg) in Assembly Buffer in a total volume of 100 μL. The mixture was immediately injected onto a 100 μL loop connected to a Superdex 200 HR 5/200 GL column running at 0.4 ml/min in Buffer AC (50 mM Tris-HCl, pH 8; 100 mM NaCl). Fractions were collected, pooled, concentrated using a 100 kDa polysulfone filter (Pall Corporation, USA), and stored at 4°C. For on-column reconstitution of FhuA, 500 μL of the protein (1 mg) was mixed with NSP (1.8 mg) in Buffer A + 0.05% LDAO, and injected onto a 500 μL loop connected to a Superdex 200 HR 10/300 GL column running at 0.5 mL/min in Buffer AC.

## 'In-gel' peptidisc reconstitution

The target membrane protein (~1.25 μg) was mixed with increasing concentrations of NSP$_r$(0–2.5 μg) and allowed to incubate for 1–2 min at room temperature. The mixture was then supplemented with Buffer A to bring the final detergent concentration below its CMC (0.008% and 0.01% for DDM and LDAO, respectively) while keeping the final volume to 15 μL. A solution of glycerol was added to 10% final to facilitate loading on 4–12% CN-PAGE. The electrophoresis was set constant at 25mA for 1 hr at room temperature. Bands were visualized by Coomassie Blue G250 staining.

### 'On-bead' peptidisc reconstitution

Crude membranes (10 mL at 7.5 mg/ml total protein content) containing overexpressed MalFGK$_2$ were solubilized in Buffer A + 1% DDM for 1 hr at 4°C before removal of insoluble aggregate by ultracentrifugation (100,000 x g, 1 hr, 4°C). The solubilized membrane proteins were incubated with 200 µl of Ni-NTA resin (Qiagen) pre-equilibriated in Buffer A + 0.02% DDM for 1 hr at 4°C. The Ni-NTA beads were collected by low-speed centrifugation (3000 x g, 3 min), washed twice with 10 CV of Buffer B supplemented with 0.02% DDM. Post-washing, 10 CV of Assembly Buffer (1 mg/mL NSP$_r$ in 20 mM Tris-HCl pH 8) was added to the beads and allowed to incubate for 5 min on ice. The Assembly Buffer was removed and the beads loaded into a gravity column with 10 CV of Buffer B (50 mM Tris-HCl, pH 8; 200 mM NaCl, 10% glycerol, 15 mM imidazole). The assembled peptidiscs were subsequently treated with 500 µL of Buffer C (50 mM Tris-HCl, pH 8; 100 mM NaCl;10% glycerol; 400 mM imidazole) to elute the peptidisc from the affinity resin. The same procedure was done in parallel, except the NSP$_r$ was omitted from the Assembly Buffer and 0.02% DDM was included in Buffer A, B, and C.

### Reconstitution of the BRC in Peptidiscs, low lipid nanodiscs, and styrene maleic acid nanoparticles

The purified BRC complex (1 mg/mL) was mixed at a 1:1.8 (µg/µg) ratio with NSP$_r$ followed by 10-fold dilution in Buffer A to decrease the LDAO concentration to 0.003%. For formation of low-lipid nanodiscs, the purified BRC complex was instead mixed at a 1:2 (mol/mol) ratio with MSP1D1 before dilution. Alternatively, an equivalent amount of BRC was diluted in Buffer A supplemented with 0.03% LDAO, 0.02% DDM, 0.1% SMA or 0.1% SDS as described. After incubation for 10 min on ice, aggregated proteins were removed by centrifugation (13,000 x g, 10 min at 4°C). Peptidisc formation was confirmed by analysis on CN-PAGE.

### Reconstitution of MalFGK$_2$ and BRC in proteoliposomes

Proteoliposomes were prepared at a molar protein:lipid ratio of 1:2000. Total *E. coli* lipids were dissolved in chloroform, dried under nitrogen and resuspended in Buffer A + 0.8% β-OG. Purified MalFGK$_2$ was added to the solubilized lipids, and the detergent was removed by overnight incubation at 4°C with Amberlite XAD-2 adsorbent beads (Supelco). The proteoliposomes were isolated by ultracentrifugation (100,000 × g, 60 min at 4°C) and resuspended in 20 mM Tris–HCl, pH 8 before use in ATPase assays. The same procedure was employed for the BRC, but a lipid mixture of DOPC: DOPG (80:20 mol/mol) was utilized in place of total *E.coli* lipids.

### Native gel electrophoresis

Equal volumes of 4% and 12% acrylamide solutions were prepared in advance (*Supplementary file 2*). Linear gradient gels were formed by gradual mixing of the two solutions (35 mL each) at a flow rate of 2 ml/min using a 100 mL gradient mixer (Sigma). The cross-linking agents, TEMED and ammonium persulfate, were added immediately before gradient mixing. Once poured, plastic wells (Biorad) were inserted and gels allowed to cure for 90 min before storage at 4°C. For clear-native PAGE, anode and cathode buffers consisted of Buffer N (37 mM Tris-HCl; 35 mM Glycine; pH 8.8). For blue-native PAGE, anode buffer consisted of Buffer N + 180 µM Coomassie Blue G-250, and cathode buffer contained Buffer N only.

### Dynamic and static light scattering analysis

Aliquots of MalFGK$_2$-NSPr were analyzed by static light scattering. Static light scattering analysis were performed using a WTC-050S5 column (Wyatt Technologies) connected to a miniDAWN light scattering detector and interferometry refractometer (Wyatt Technologies). Data were recorded in real time and the molecular masses were calculated using the Debye fit method using the ASTRA software (Wyatt Technology).

### Sample preparation and EM image acquisition

MalFGK$_2$-peptidisc sample (0.035 mg/mL) reconstituted by on-column method was applied onto negatively glow-discharged carbon-coated grids (400 mesh, copper grid) for 1 min, and excess liquid was removed by blotting with filter paper. Freshly prepared 1.5% uranyl formate (pH 5) was added

(5 µl) for 1 min and then blotted. Around 200 digital micrographs were collected using a FEI Tecnai G2 F20 microscope operated at 200 kV and equipped with a Gatan Ultrascan 4k × 4 k Digital CCD Camera. The images were recorded at defocus between 0.7 and 1.4 µm at a magnification of 67,000X at the camera and a pixel size of 2.24 Å.

## EM data processing and image analysis

Contrast transfer function parameters were determined using CTFFIND3 (*Mindell and Grigorieff, 2003*). We selected 31188 protein particles using e2boxer from the EMAN2 software suite (*Tang et al., 2007*) and extracted with a box size of 96 × 96 pixels. Particles were classified using a likelihood 2D classification with 16 seeds with the RELION-1.3 software suite (*Scheres, 2012*). A 2D variance of particles contained in side view (3175 particles) was computed with SPARX to estimate the peptidisc diameter variation (*Hohn et al., 2007*). The measurements were done using e2display from the EMAN2 software suite (*Corin et al., 2011*) on the side views shown in *Figure 2C*.

## FhuA binding assay

FhuA-$MSP_{L156}$ nanodiscs were prepared as previously described (*Lee et al., 2016*). FhuA-$NSP_r$ was prepared by on-column peptidisc reconstitution. About 2 µg of FhuA reconstituted into either $MSP_{L156}$ or $NSP_r$ was incubated with $TonB_{23-329}$ (2 µg) or ColM (5 µg) in the presence or absence of ferricrocin for 5 min at room temperature. The protein complexes were separated by CN-PAGE and visualized by Coomassie blue staining. Neither monomeric TonB nor ColM migrate on CN-PAGE due to their isoelectric points > pH 8.8.

## Mass spectrometry

BRC peptidisc, $MalFGK_2$ peptidisc, and FhuA peptidisc were prepared by 'on-column' reconstitution in 100 mM ammonium acetate, pH 7.0 at protein to $NSP_r$ (g/g) ratios of 1:1.8, 1:1.6, and 1:1.8, respectively. Mass spectrometry measurements were performed in positive ion mode on a Synapt G2S quadrupole-ion mobility separation-time-of-flight (Q-IMS-TOF) mass spectrometer (Waters, Manchester, UK) with a nanoflow electrospray ionization ESI (nanoESI) source. Borosilicate capillaries (1.0 mm o.d., 0.68 mm i.d.) were pulled in-house using a P-1000 micropipette puller (Sutter Instruments, Novato, CA). A voltage of ~1.0 kV was applied to a platinum wire was inserted into the nano-ESI tip. A source temperature of 60°C and a Cone voltage of 30 V were used. Argon was used in the Trap and Transfer ion guides, at pressures of $2.77 \times 10^{-2}$ mbar and $2.84 \times 10^{-2}$ mbar, respectively, and the Trap and Transfer voltages were 5 V and 2 V, respectively. All data were processed using MassLynx software (*v*4.1). Spectral deconvolution was performed with the UniDec (*Marty et al., 2015*) deconvolution algorithm using the following parameters: *m/z* range – 7000 to 9500 ($MalFGK_2$ peptidisc), 5500 to 9000 (BRC peptidisc), 5000 to 10000 (FhuA peptidisc); Subtract minimum - 50.0; Gaussian Smoothing - 10.0; Bin every 1.0; Linear *m/z* (constant delta *m/z*); Charge Range - 20 to 40 ($MalFGK_2$ peptidisc), 10 to 30 (BRC peptidisc), 10 to 30 (FhuA peptidisc); Mass range - 200,000 to 300,000 ($MalFGK_2$ peptidisc), 100,000 to 180,000 (BRC peptidisc), 100,000 to 170,000 (FhuA peptidisc); Sample Mass Every 1.0 Da; Peak FWHM (Th) 4.0; Peak Shape Function - Gaussian; Charge Smooth Window - 1.0; Mass Difference - 4474.0; Mass Smooth Window - 1.0; Maximum number of iterations - 1000. Spectral files were loaded as text files containing intensity and *m/z* values.

## Absorbance spectroscopy

Absorption spectra were recorded using a Hitachi U-3010 spectrophotometer. A blank measurement was recorded in Buffer A (+0.03% LDAO for detergent purified BRC). Samples were incubated in a PCR thermocycler at the indicated temperature, and then measured at the desired time points in a quartz cuvette at room temperature. Spectra were collected between 600 nm and 1100 nm (scan time ~20 s) at intervals of 1.5 min. For comparisons of spectra between conditions, spectra were normalized to a value of 1.0 at 804 nm.

## Fluorescence measurements

The BRC complex into the indicated detergent or reconstituted into peptidiscs was incubated at varying temperatures in a PCR thermocycler for 5 min, then 3 µL of the mixture dotted onto

nitrocellulose paper pre-wetted in Buffer A. The dot blot was imaged using a LICOR odyssey infrared fluorescence scanner (excitation 680 nm, emission 700 nm). Fluorescence intensity was quantified by Image J.

## NSP$_r$ quantification

The MalFGK$_2$ and BRC peptidiscs were prepared by on-column reconstitution on a Superdex 5/25 column equilibrated in Buffer A, followed by one additional gel filtration step to ensure full removal of free NSP$_r$. MalFGK$_2$ (1 µg), FhuA (2 µg), and BRC (2 µg) peptidiscs were analyzed by 15% SDS-PAGE. Gels were stained with Coomassie Blue G-250, and destained overnight before fluorescence measurement (excitation 680 nm, emission 700 nm) on a LICOR Odyssey scanner. The band corresponding to the NSP$_r$ peptide was quantified by densitometry using Image J and compared to a standard curve of NSP$_r$(0–2 µg) loaded on the same gel. The determined NSP$_r$ amount was then subtracted from the total amount of protein loaded on the gel to determine the amount of reconstituted membrane protein in the peptidisc. Membrane protein content in peptidisc (g) = total protein in peptidisc (g) - measured NSP$_r$ content (g). We used these calculated mass measurements and the molecular weight (MW) for NSP$_r$ (4.5 kDa), MalFGK$_2$ (173 kDa), FhuA (80 kDa) and BRC (94 kDa) to calculate NSP$_r$ stoichiometry as follows;

$$\text{NSP}_r\,\text{Stoichiometry} = \frac{\text{MW Membrane protein (g/mol)}}{\text{MW NSP}_r\,\text{(g/mol)}} \times \frac{\text{Measured NSP}_r\,\text{content (g)}}{\text{Membrane protein content in peptidisc (g)}}$$

Each experiment was repeated in triplicate on three different gels. We note that detergent-purified FhuA co-purified with a contaminant, thought to be short chain lipopolysaccharides, that migrated to the same position as NSP$_r$, therefore FhuA was reconstituted using NSP$_r$ labelled with a biotin group (NSP$_{rbio}$). To quantify NSP$_{rbio}$, western blots were incubated with streptavidin conjugated to Alexafluor 680 in phosphate buffered saline (PBS), followed by several washes in PBS + 0.1% Tween. Western blots were imaged on a LICOR Odyssey scanner fluorescence (excitation 680 nm, emission 700 nm), and the bands corresponding to NSP$_{rbio}$ quantified in Image J.

## Lipid extraction and quantification

The MaFGK$_2$ and BRC peptidiscs were prepared on-bead, and the FhuA peptidisc was prepared on-column. MalFGK$_2$ (40 µg), FhuA (40 µg), and BRC (80 µg) peptidiscs were diluted to a final volume of 200 µL of Buffer A, then mixed with 800 µL of a 2:1 solution of methanol:chloroform for 10 min at 25°C in glass screw cap vials. 200 µL of chloroform and 200 µL of distilled water were added sequentially, vortexed briefly, and the resulting two phase system separated by low-speed centrifugation (3000 r.p.m., 10 min). The organic phase was dried under nitrogen, and stored at −20°C. Total phosphate content was determined by a modified version of the malachite green assay (*Lanzetta et al., 1979*). Malachite green reagent was prepared as follows: ammonium molybdate (4.2 g) was dissolved in 100 mL of 4M HCl, then mixed with 300 mL malachite green (135 mg) dissolved in distilled water. The solution was mixed for 1 hr at 4°C, filtered, and stored at 4°C before use. Dried lipid extracts were subsequently incubated with 1 mL of 70% perchloric acid for 3 hr at 130°C, and then 20 µL of the resulting solution mixed with 500 µL of the malachite green reagent for 5 min at room temperature before absorbance measurement at 660 nm. Phosphate standards (KH$_2$PO$_4$) were diluted into perchloric acid and used to prepare a standard curve with phosphate concentrations ranging from 0.01 nmol to 1 nmol PO$_4$. For thin layer chromatography (TLC) analysis, dried lipids were resuspended in 30 µL of chloroform, and 10 µL were dotted onto a TLC Silica gel 60 (Millipore). The TLC was developed in a solution of 35:25:3:28 chlorofrom:triethylamine:$_d$H$_2$O:ethanol. Plates were dried in an oven for 5 min at 150°C. Lipids were visualized by lightly wetting plates in a solution of 10% Cu$_2$S0$_4$ in 8.5% phosphoric acid, followed by heating for 5 min at 150°C.

## Other methods

The MalFGK$_2$ ATPase activity was determined by monitoring the release of inorganic phosphate using the malachite green method (*Lanzetta et al., 1979*). Protein and peptide concentrations were determined by Bradford assay (*Prehna et al., 2012*). SMA polymer containing 2:1 styrene to maleic acid ratio was prepared following the procedure described by *Dörr et al. (2014)*. In brief, 10% of SMA 2000 (Cray Valley), was refluxed for 3 hr at 80°C in 1M KOH, resulting in complete solubilization

of the polymer. Polymer was then precipitated by dropwise addition of 6M HCl accompanied by stirring and pelleted by centrifugation (1500 x g for 5 min). The pellet was then washed 3 times with 50 mL of 25 mM HCl, followed by a third wash in ultrapure water and subsequent lyophilization. Lyophilized SMA was later re-suspended at 10% wt/vol in 25 mM Tris-HCl, and the pH of the solution adjusted to 8 with 1M NaOH. Peptide hydrophobic moment and electropotential was calculated using the 3D-HM calculator (*Reißer et al., 2014*). Sequences corresponding to NSP or NSP$_r$ were calculated with the C-terminus specified as (COO$^-$) and N-terminus specified as (NH$_3^+$). UV absorbance of solubilized peptides was measured by Nanodrop.

## Acknowledgements

This work was supported by operating grants from the Canadian Institutes of Health Research (74525MOP to FD), the Natural Sciences and Engineering Research Council of Canada (Discovery Grant 2796 to JTB), and Genome British Columbia (SOF153 to JTB).

## Additional information

### Competing interests

Franck Duong: Has opened a website to distribute peptides to the academic community, registered the Peptidisc term and filed a provisional patent via the University of British Columbia. The other authors declare that no competing interests exist.

### Funding

| Funder | Grant reference number | Author |
|---|---|---|
| Canadian Institutes of Health Research | 74525MOP | Franck Van Hoa Duong |
| Natural Sciences and Engineering Research Council of Canada | Discovery Grant 2796 | J Thomas Beatty |
| Genome British Columbia | SOF153 | J Thomas Beatty |

The funders had no role in study design, data collection and interpretation, or the decision to submit the work for publication.

### Author contributions

Michael Luke Carlson, Conceptualization, Data curation, Formal analysis, Validation, Investigation, Visualization, Methodology, Writing—original draft, Writing—review and editing, Conceived the study, Contributed to discussion and writing, Performed light scattering experiments, Did all sample preparation and data analysis unless otherwise stated; John William Young, Conceptualization, Methodology, Writing—review and editing, Prepared detergent purified SecEYG, Conceived the study, Contributed to discussion and writing; Zhiyu Zhao, Investigation; Lucien Fabre, Data curation, Formal analysis, Investigation, Writing—original draft, Prepared and analyzed electron microscopy data; Daniel Jun, Resources, Data curation, Investigation, Writing—review and editing, Performed BRC stability experiments; Jianing Li, Formal analysis, Investigation, Visualization; Jun Li, Data curation, Formal analysis, Investigation, Visualization, Writing—original draft, Performed and analyzed intact native MS experiments; Harveer Singh Dhupar, Investigation, Prepared samples for analysis by mass spectrometry; Irvin Wason, Investigation, Performed peptide titrations, Analyzed and presented the data; Allan T Mills, Investigation, Prepared detergent purified OmpF; J Thomas Beatty, Resources, Supervision, Funding acquisition, Investigation, Visualization, Writing—original draft, Writing—review and editing; John S Klassen, Resources, Software, Supervision, Funding acquisition, Investigation, Visualization, Writing—original draft, Writing—review and editing; Isabelle Rouiller, Franck Duong, Conceptualization, Resources, Software, Formal analysis, Supervision, Funding acquisition, Validation, Investigation, Visualization, Methodology, Writing—original draft, Project administration, Writing—review and editing

## Author ORCIDs

Michael Luke Carlson (iD) https://orcid.org/0000-0002-3807-6516

John S Klassen (iD) https://orcid.org/0000-0002-3389-7112

Franck Duong (iD) https://orcid.org/0000-0001-7328-6124

## Decision letter and Author response

Decision letter https://doi.org/10.7554/eLife.34085.022

Author response https://doi.org/10.7554/eLife.34085.023

# Additional files

## Supplementary files

• Supplementary file 1. Amphipathic scaffolds used in this study. The length of the scaffold proteins was calculated by multiplying the number of amino acids (aa) by 1.5 Å, which is the rise given by an amino acid structured in an alpha-helix. For the MSPs scaffolds, the number of amino acids was from the TEV cleavage site (ENYLFQ//GXXX) to the C-terminus of the proteins.
DOI: https://doi.org/10.7554/eLife.34085.017

• Supplementary file 2. Native gel buffer recipes.
DOI: https://doi.org/10.7554/eLife.34085.018

• Supplementary file 3. Protein molecular weights.
DOI: https://doi.org/10.7554/eLife.34085.019

• Transparent reporting form
DOI: https://doi.org/10.7554/eLife.34085.020

## Data availability

All data generated or analysed during this study are included in the manuscript and supporting files.

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
