## [Decision Letter]

[Editors’ note: this article was originally rejected after discussions between the reviewers, but the authors were invited to resubmit after an appeal against the decision.]

Thank you for submitting your work entitled "The Peptidisc, a Simple Method for Stabilizing Membrane Proteins in Detergent-free Solution" for consideration by *eLife*. Your article has been evaluated by a Senior Editor and three reviewers, one of whom, Volker Dötsch (Reviewer #1), is a member of our Board of Reviewing Editors. The following individual involved in review of your submission has agreed to reveal their identity: Markus A Seeger (Reviewer #3).

Our decision has been reached after consultation between the reviewers. Based on these discussions and the individual reviews below, we regret to inform you that your work will not be considered further for publication in *eLife*.

While all reviewers see the potential usefulness and impact of the peptidiscs it seems that the use of peptides to solubilize membrane proteins has been proposed and implemented before. As you will see in the individual reviews the lack of comparison to these existing methods has been criticized. In addition, a more detailed analysis of how the peptides achieve solubilization is lacking in comparison to published peptide based methods. And finally, a real proof that peptidiscs are superior to existing methods for structure determination as claimed would have been beneficial. Overall, the manuscript describes a very interesting alternative to existing methods to solubilize membrane proteins, it seems however less evident that the method will provide a major breakthrough.

*Reviewer #1:*

Carlson et al. describe a peptide based system that keeps membrane proteins soluble with tightly attached lipid molecules. In contrast to the well-established nanodisc system this system is based on a variable number of peptides that surround the membrane protein and stabilize it. No addition of lipids is necessary and the high flexibility of the system does not make the engineering of differently long peptides as scaffolds necessary. They show with five different membrane proteins that they can be stably solubilized in these discs and provide a very careful assessment of the peptide/protein/lipid composition of these discs. They also show that these discs can be formed in several different ways starting with detergent solubilized membrane proteins.

This is an interesting system that can provide an attractive alternative to the existing hydrophobic systems to solubilize membrane proteins. The study is well documented and carefully performed.

1) The proteins used are relatively large membrane protein complexes or b-barrel proteins that are stable even in detergent micelles. For those membrane proteins the peptidisc will be a good alternative. However, for membrane proteins that are not stable in detergent micelles, are expressed cell-free or resolubilized from inclusion bodies it is not clear if they can be stabilized with these discs. A GPCR or a transporter would have been good alternatives.

2) One of the advantages over nanodiscs that the authors advertise is that the peptidiscs can be used for structural investigations. It would have been good to show for example an NMR spectrum of a membrane protein in such a peptidisc and to show that this is advantages relative to larger nanodiscs.

*Reviewer #2:*

The manuscript by Carlson et al. includes a considerable amount of data on five different membrane proteins solubilized into bi-helical peptides. This is combined with three reconstitution methods to transfer proteins into peptidiscs on columns, beads or gels. As such, it potentially provides a useful methodological advance and benchmark within the rapidly developing field of preparation and analysis of membrane proteins in nanometer sized discs.

However, there are overstatements and lack of clarity of key points that need to be addressed.

The Abstract states that "the peptidisc just requires a short amphipathic bi-helical peptide (NSP) and no extra lipids". However, the proteins are initially prepared using detergents and are expressed in *E. coli*. Hence recombinant expression in for e.g. *E. coli* and detergent appear to be also required.

The authors go on to say that "This drawback has led researchers to develop detergent-free alternatives such as amphipols,[Popot, 2010] SMALPs,[Lee et al., 2016] saposin-lipoparticles[Frauenfeld et al., 2016] and the popular nanodisc system.[Bayburt, Grinkova and Sligar, 2004; Denisov et al., 2004]" Again, some of these methods also require detergent to be added. This is a distinction that needs to be clarified. The statement "We present here the peptidisc as a simple assembly method to support membrane protein in detergent-free solution" similarly needs to be corrected as it is not clear that the methods are simple or detergent-free. The concluding statement in the Introduction that "we show that the NSP peptide may well be the universal scaffold for stabilizing both α-helical and β-barrel membrane proteins of different size, topology, and complexity" and following the discussion that "These advantages combined suggest that the peptidisc should diminish the challenges associated with biochemical, structural and pharmacological characterization of membrane proteins, making the peptidisc an efficient and perhaps universal tool for stabilizing these proteins in membrane- and detergent-free solution" are overstated. For this to be a useful paper, the authors need to indicate the limitations of the method. In the Materials and methods used for on column peptidisc reconstitution detergents such as LDAO are used. Doesn't the peptide itself denature in such detergents, and would this not limit the effectiveness of this reagent?

The yield, purity and activity should be given quantitatively for each protein in a peptidisc vs. in detergent alone, and ideally also vs. in a liposome. This would allow the method to be objectively compared. Do the proposed methods not require detergent that could strip away natively bound lipids and destabilize membrane proteins? Is this not a limitation? Ideally assays should be given across a temperature range to ensure that folded protein is being measured for activity.

The identities of bound lipids should also be stated, rather than non-specific statements about lipid content like "Also, because lipids (i.e. annular lipids) can remain tightly bound to membrane proteins during purification [Bechara et al., 2015], we also determined the lipid content by thin layer chromatography and photocolorimetric methods" and "Following the same approach applied to MalFGK_2_ above, we quantified the individual peptide and lipid components of the FhuA peptidisc (Figure 3B and A), resulting in an average of 8 ± 3 phospholipids and 10 ± 2 NSP per FhuA peptidisc". The low number of lipid molecules present (4 and 8 in the cases of BRC and FhuA) indicates that only the most tightly bound lipids remain in the peptidisc, and that a disc shape cannot be assumed.

The presence of an apparently non-physiological multimer (Sec(EYG)n) in Figure 2D is glossed over. This needs to be explained as it indicates that use of NSP is leading to potentially artefactual multimeric states.

In Figure 2F, why do proteins with different molecular weight share the same RR50? Would one not expect to see a higher number of peptides interacting with larger assembly or with different membranes or cell types? The authors didn't explain their perspective on this, nor is it clear how this was optimized, despite being a significant cost and determinant of success of the method. A recommended molar concentration of peptide for reconstitution should be indicated and justified.

The description of the peptide being like a belt around the protein and of the disc shape of peptidiscs needs to be justified with experimental data and/or references. There is one set of negative stain EM data of MalFGK_2_ in peptidiscs showing the presence of a number of pairs and triplets of discs. Is this not significant and indicative of disc-disc interactions, perhaps mediated by NSP peptides? If so shouldn't lower peptide concentrations be used to minimize such stacks in biophysical assays? Also, the scale bar in the bottom panel of Figure 1C appears to indicate 50 (not 5) Å. If so this is inconsistent.

*Reviewer #3:*

The manuscript by Carlson et al. describes a protocol for membrane protein stabilization using an amphiphilic 37 amino acid peptide called nanodisc scaffold protein (NSP) to result in a peptidisc, which corresponds to the membrane protein surrounded by NSP and annular lipids. The method is applied and validated by reconstituting an ABC importer (maltose transporter), two outer membrane proteins, SecEYG and a bacterial reaction center (BRC). The ratio between protein and NSP peptide as well as the number of bound phospholipids was determined using a set of different biophysical methods. The ATPase activity of the maltose transporter embedded in a peptidisc was found to be strongly coupled to the presence of the maltose binding protein MalE, which is a hallmark of samples reconstituted in proteoliposomes or nanodiscs, whereas very poor coupling is observed in detergent solution. For BRC, the peptidisc was shown to stabilize the membrane protein as compared to the (rather harsh) detergent LDAO.

The peptidisc is proposed in the manuscript to be equivalent to other membrane reconstitution methods such as the widely used nanodisc system and the classical proteoliposomes with regard to conformational coupling (shown for the maltose transporter) as well as membrane protein stabilization. On the other hand, the NSP peptide is used as a surrogate for a detergent, i.e. it is very simple to use and just needs to be added to a solubilized membrane protein to replace the initial detergent as is often done with short chain detergents replacing for example DDM, which is widely used to solubilize membrane proteins.

The manuscript appears solid with regard to the biophysical analyses of the peptidisc complexes and their content of NSP peptides and annular lipids. Nevertheless, I am a bit skeptical whether a peptidisc represents a reconstitution in the classical sense, because the NSP peptides directly interact with the transmembrane helices (driven by hydrophobic protein-protein interactions mediated by amino acid side chains) instead of lipids via their aliphatic fatty acid chains. Nevertheless, the method (although not being a novel concept at as outlined below) seems to be rather easy and versatile in its application.

1) As a potential future user (and a current user of the nanodisc method), it is critical to know how expensive the synthesis (or any alternative preparation) of the NSP peptide is. I assume it is more expensive than detergents, but in contrast to the MSP protein used for nanodiscs, it might be less expensive. Along the same lines, the authors stated that the purity of their chemically synthetized NSP peptide was "more than 80%". Is this sufficient? What are the impurities. Is this rather low purity linked to the price of the synthesis?

2) How aggregation-prone are peptidiscs? The authors hypothesize that the NSP peptides build a regular belt around the membrane protein, akin to the MSP belt. But is this really feasible? It is clear that the peptides arrange such that the hydrophobic parts of the membrane protein are covered by its hydrophobic side, while the hydrophilic face of the peptides remains exposed to the solvent. But this process is likely to be stochastic in the sense that there remain hydrophobic gaps on the membrane protein surface, as well as on some of the not perfectly placed NSP peptides. These remaining hydrophobic surfaces would then serve as nucleation points for protein aggregation. Remaining detergent molecules originating from the solubilization and purification of the membrane protein may shield these remaining hydrophobic surfaces.

3) Related to the above comment: Do peptidiscs still contain detergents? It is well known that quantitative detergent exchange is not a trivial endeavor. Often the initial detergent used to solubilize the membrane protein remains present to some degree. The authors need to show, whether in the peptidiscs there are still some detergents remaining (next to annular lipids and the NSP peptide).

4) How big is the NSP belt compared to detergent belts of different sizes? This question is highly relevant for membrane protein crystallization. The peptidisc appears more compact than a nanodisc and membrane proteins in peptidiscs might be crystallized. Did the authors try to crystallize their five membrane proteins used in this study, for which high resolution crystal structures (determined in detergent) exist?

5) Does the peptidisc really mimic the natural environment of a membrane protein? The authors carefully worked out that the peptidisc contains annular lipids. However, annular lipids are also contained in detergent-purified samples. The authors should address this question by comparing the content of annular lipids side-by-side and over time using the same transporter purified in detergent as well as in peptidiscs and measuring by mass spectrometry the mass (or loss of mass over time) of the entire complex.

6) The manuscript lacks a proper Discussion. Most problematically, the authors do not discuss at all that the concept of peptides used as detergent surrogates is not novel at all. Rather, it was first described in the 80s and 90s as peptitergent (you can find this word even on Wikipedia). See also in Schafmeister CE, Miercke LJ, Stroud RM. Structure at 2.5 A of a designed peptide that maintains solubility of membrane proteins. Science. 1993 Oct 29;262(5134):734-8. In addition, there are lipopeptide detergents (LPDs), nano-structured β-sheets (see Tao H, Lee SC, Moeller A, Roy RS, Siu FY, Zimmermann J, Stevens RC, Potter CS, Carragher B, Zhang Q. Engineered nanostructured β-sheet peptides protect membrane proteins. Nat Methods. 2013 Aug;10(8):759-61). Further, there was the development of short peptides for membrane stabilization, which are much cheaper to produce than the 37 aa NSP peptide (see Kiley P, Zhao X, Vaughn M, Baldo MA, Bruce BD, Zhang S. Self-assembling peptide detergents stabilize isolated photosystem I on a dry surface for an extended time. PLoS Biol. 2005 Jul;3(7):e230. Epub 2005 Jun 21.)

Important is the question whether at all or to what extent the NSP peptide is superior to these previously described methods using amphiphilic peptides, in particular the short designer peptides as described in Kiley et al., 2005). Critically, this point needs to be addressed experimentally by a functional assay including the maltose transporter and/or the BRC.

The Discussion should also include the concept of membrane protein stabilization via peptide-protein interactions (peptidisc) versus lipid protein interactions (nanodisc/saposins/proteoliposomes). And finally, it needs to address the amphipols (which are conceptually most closely linked to peptitergents and peptidiscs) in more detail.

[Editors’ note: what now follows is the decision letter after the authors submitted for further consideration.]

Thank you for resubmitting your work entitled "The Peptidisc, a Simple Method for Stabilizing Membrane Proteins in Detergent-free Solution" for further consideration at *eLife*. Your revised article has been favorably evaluated by Richard Aldrich (Senior Editor), a Reviewing Editor, and two reviewers.

The manuscript has been improved but there are some remaining issues that need to be addressed before acceptance, as outlined below:

1) What is interesting in the revised version is the fact that the sequence of the original NSP peptide as it is present in ApoA1 was in fact reversed and that this resulted in an increase of solubility of the peptide. Why was this information completely lacking in the first version?

2) The exact sequence of the NSPr peptide should be provided directly in the Materials and methods section under section "Biological reagents and peptides" (I had difficulties finding it in Supplementary file 1).

3) The authors are now stating in the manuscript that the method is inexpensive. However, I would still be interested to know how much a mg or a gram costs and how much you will need for an experiment (by providing a reasonable range).

4) Concerning the impurities of the peptide, the authors gave an interesting answer in the response letter (i.e. that the great majority of the impurity is in fact the same peptide missing its final amino acid). This should be mentioned in the manuscript as well.

5) The authors refer to a homepage to order/obtain the peptide: www.peptidisc.com. However, the homepage does work yet. It seems that the authors will or have already founded a start-up company to commercialize the peptidisc. Potential financial competing interests have to be declared.

---

## [Author Response]

[Editors’ note: the author responses to the first round of peer review follow.]

Reviewer #1:[…] 1) The proteins used are relatively large membrane protein complexes or b-barrel proteins that are stable even in detergent micelles. For those membrane proteins the peptidisc will be a good alternative. However, for membrane proteins that are not stable in detergent micelles, are expressed cell-free or resolubilized from inclusion bodies it is not clear if they can be stabilized with these discs. A GPCR or a transporter would have been good alternatives.

We agree with the reviewer, the peptidisc like other mimitics will not work with proteins that are hard to purify or too unstable in detergent. However, in the case of the BRC complex, we observe 100 fold stability increase in peptidic versus LDAO detergent at elevated temperature. The immediate aim of our work is to present the peptidisc as a new tool, so that it can be tested by other researchers on their favorite target. For example, current biochemical work on GPCRs involves amphipols, and not so much nanodiscs perhaps due to the required addition of exogenous lipids which complicate the reconstitution. The peptidisc therefore has also potential to replace amphipols which are inherently polydisperse. Further studies characterizing the peptidisc’s usefulness for cell-free expression systems is certainly of interest, however beyond the scope of this introductory article. We feel that a cell-free expression should be investigated as a separate study with all the different possible membrane mimetics (nanodiscs/SMA/peptergents etc.) to be truly useful.

2) One of the advantages over nanodiscs that the authors advertise is that the peptidiscs can be used for structural investigations. It would have been good to show for example an NMR spectrum of a membrane protein in such a peptidisc and to show that this is advantages relative to larger nanodiscs.

We only mentioned that peptidisc, given their compositional homogeneity, should be advantageous for structural studies. Our laboratory is however not equipped for such structural analysis but we are actively seeking collaborators.

Reviewer #2:The manuscript by Carlson et al. includes a considerable amount of data on five different membrane proteins solubilized into bi-helical peptides. This is combined with three reconstitution methods to transfer proteins into peptidiscs on columns, beads or gels. As such, it potentially provides a useful methodological advance and benchmark within the rapidly developing field of preparation and analysis of membrane proteins in nanometer sized discs. However, there are overstatements and lack of clarity of key points that need to be addressed.The Abstract states that "the peptidisc just requires a short amphipathic bi-helical peptide (NSP) and no extra lipids". However, the proteins are initially prepared using detergents and are expressed in E. coli. Hence recombinant expression in for e.g. E. coli and detergent appear to be also required.

To prevent further confusion, we have modified this statement to “Reconstitution of a detergent solubilized membrane protein into a peptidisc only requires a short,”

The authors go on to say that "This drawback has led researchers to develop detergent-free alternatives such as amphipols,[Popot, 2010] SMALPs,[Lee et al., 2016] saposin-lipoparticles[Frauenfeld et al., 2016] and the popular nanodisc system.[Bayburt, Grinkova and Sligar, 2004; Denisov et al., 2004]" Again, some of these methods also require detergent to be added. This is a distinction that needs to be clarified.

We have introduced a new section in the Introduction to present and to explain the main differences between synthetic scaffolds and the other detergent free alternatives.

The statement "We present here the peptidisc as a simple assembly method to support membrane protein in detergent-free solution" similarly needs to be corrected as it is not clear that the methods are simple or detergent-free.

It is both. We have rephrased this statement.

The concluding statement in the Introduction that "we show that the NSP peptide may well be the universal scaffold for stabilizing both α-helical and β-barrel membrane proteins of different size, topology, and complexity" and following the discussion that "These advantages combined suggest that the peptidisc should diminish the challenges associated with biochemical, structural and pharmacological characterization of membrane proteins, making the peptidisc an efficient and perhaps universal tool for stabilizing these proteins in membrane- and detergent-free solution" are overstated. For this to be a useful paper, the authors need to indicate the limitations of the method.

The initial Discussion was very short because we initially submitted this paper as a method. We now have completely revisited the Discussion to highlight the pro and con of the various scaffolds and where other methods may be more appropriate. We have also increased the Introduction to present those other methods and why the peptidisc is evidently superior on certain specific aspects.

In the Materials and methods used for on column peptidisc reconstitution detergents such as LDAO are used. Doesn't the peptide itself denature in such detergents, and would this not limit the effectiveness of this reagent?

Reconstitution in peptidisc, like in nanodisc, occurs upon dilution of the detergent. This collapses the micelle, allowing the peptide to fold around the target proteins. Unfolding of the peptide is not an issue. Our data show that LDAO is not limiting the effectiveness of the method.

The yield, purity and activity should be given quantitatively for each protein in a peptidisc vs. in detergent alone, and ideally also vs. in a liposome. This would allow the method to be objectively compared. Do the proposed methods not require detergent that could strip away natively bound lipids and destabilize membrane proteins? Is this not a limitation?

We have performed a side-by-side comparison of the BRC complex reconstituted in liposomes, nanodiscs, peptidiscs, SMA polymer, and LDAO detergent and presented the results in the revised manuscript. All membrane mimetic systems show a similar increase in thermostability. Delipidation is a general problem with detergent solubilization. However, we show that the peptide increases the thermostability of the delipidated BRC complex. This is likely due to i) removal of denaturing detergents, and ii) a more stable hydrophobic environment as compared to a detergent micelle. Thus, even without extra lipids, the peptidisc is able to stabilize the BRC as much as in a proteoliposome (Figure 10—figure supplement 1). Detergent delipidation can also be a useful for increasing purity of membrane proteins. For example, during purification of FhuA, the inner membrane is first removed by solubilization in Triton X-100 before addition of LDAO to release FhuA from the outer membrane. This significantly increases the final purity.

Ideally assays should be given across a temperature range to ensure that folded protein is being measured for activity.

We have shown that MalFGK_2_ in peptidisc remains folded throughout the reconstitution using BN-PAGE and ATPase activity assays.

The identities of bound lipids should also be stated, rather than non-specific statements about lipid content like "Also, because lipids (i.e. annular lipids) can remain tightly bound to membrane proteins during purification [Bechara et al., 2015], we also determined the lipid content by thin layer chromatography and photocolorimetric methods" and "Following the same approach applied to MalFGK_2_ above, we quantified the individual peptide and lipid components of the FhuA peptidisc (Figure 3B and A), resulting in an average of 8 ± 3 phospholipids and 10 ± 2 NSP per FhuA peptidisc". The low number of lipid molecules present (4 and 8 in the cases of BRC and FhuA) indicates that only the most tightly bound lipids remain in the peptidisc, and that a disc shape cannot be assumed.

We have included data on the identities of the bound lipids to MalFGK_2_. We also show that FhuA and BRC have very low lipid content. However, we do not agree that this precludes a disc shape, as both the periplasmic and cytoplasmic faces of FhuA are still accessible to its soluble binding partners. Therefore, the bulk of the peptide must be located around the transmembrane domains of the transporter. Electron microscopy data included in the manuscript support further this. We have extended the Discussion to describe the possible mode of peptide association to target protein.

The presence of an apparently non-physiological multimer (Sec(EYG)n) in Figure 2D is glossed over. This needs to be explained as it indicates that use of NSP is leading to potentially artefactual multimeric states.

The oligomeric propensity of SecEYG have been reported by many researchers in the past. This is not an artifact and we have referenced this in the manuscript. Our data show that the peptidisc can capture these oligomers.

In Figure 2F, why do proteins with different molecular weight share the same RR50? Would one not expect to see a higher number of peptides interacting with larger assembly or with different membranes or cell types? The authors didn't explain their perspective on this, nor is it clear how this was optimized, despite being a significant cost and determinant of success of the method. A recommended molar concentration of peptide for reconstitution should be indicated and justified.

We have discussed this point in more detail in the Discussion, and propose a tilted conformation of the peptide to account for this interesting observation. We have also included a recommended molar concentration of peptide that is justified by the in-gel reconstitutions experiments.

The description of the peptide being like a belt around the protein and of the disc shape of peptidiscs needs to be justified with experimental data and/or references.

We now have an extended discussion on possible orientations of the peptide in the Discussion. The disc shape is clearly visible in our negative stain EM experiments.

There is one set of negative stain EM data of MalFGK_2_ in peptidiscs showing the presence of a number of pairs and triplets of discs. Is this not significant and indicative of disc-disc interactions, perhaps mediated by NSP peptides? If so shouldn't lower peptide concentrations be used to minimize such stacks in biophysical assays?

The MalFGK_2_ peptidisc preparation are monodisperse, please refer to the negative stain analysis Figure 1C. We are unclear where the reviewer is seeing disc stacking?

*Also, the scale bar in the bottom panel of Figure 1C appears to indicate 50 (not 5)* Å*. If so this is inconsistent.*

Figure 2C shows class average of a single nanodisc. 10 Å = 1 nanometre. The disc containing MalFGK_2_ is approximately 11-12nm across. The scale bar is therefore correct and consistent with stated measurements in the text.

Reviewer #3:[…] The manuscript appears solid with regard to the biophysical analyses of the peptidisc complexes and their content of NSP peptides and annular lipids. Nevertheless, I am a bit skeptical whether a peptidisc represents a reconstitution in the classical sense, because the NSP peptides directly interact with the transmembrane helices (driven by hydrophobic protein-protein interactions mediated by amino acid side chains) instead of lipids via their aliphatic fatty acid chains. Nevertheless, the method (although not being a novel concept at as outlined below) seems to be rather easy and versatile in its application.

1) Interaction – we have included a full discussion on the possible mode of interaction of the peptide with the target membrane protein. Importantly, we show that the peptidisc represents a “true” reconstitution because it is entirely stable upon removal from excess peptide, in much the same manner as a nanodisc or amphipols. This is in contrast to other peptide scaffolds (lipopeptides and peptergents), which must be maintained in buffer above their CMC to maintain protein solubility.

2) Novelty – we have carefully addressed the issue of novelty in the section Introduction and Discussion by including comparisons (pro and con) with other peptides scaffolds, protein scaffolds and synthetic scaffolds. Although the concept of trapping membrane proteins with a scaffold is not novel, there isn’t to our knowledge other reports showing functional thermostable reconstitution of protein in peptidiscs.

1) As a potential future user (and a current user of the nanodisc method), it is critical to know how expensive the synthesis (or any alternative preparation) of the NSP peptide is. I assume it is more expensive than detergents, but in contrast to the MSP protein used for nanodiscs, it might be less expensive. Along the same lines, the authors stated that the purity of their chemically synthetized NSP peptide was "more than 80%". Is this sufficient? What are the impurities. Is this rather low purity linked to the price of the synthesis?

Even low-purity peptides are still “cleaner” compared to other MSPs membrane protein scaffolds which are contaminated with lipids and other proteins due to recombinant cell protein expression (a major issue for the biotech industry when generating antibodies against membrane proteins stabilized in nanodiscs). In peptidisc, the bulk of “contaminants” consists of the NSP_r_ missing its final amino acid (aspartate) during synthesis. In comparison to MSP protein, referenced from Sigma Aldrich, the peptide is less expensive. We have included comments on these points in the Discussion.

2) How aggregation-prone are peptidiscs? The authors hypothesize that the NSP peptides build a regular belt around the membrane protein, akin to the MSP belt. But is this really feasible? It is clear that the peptides arrange such that the hydrophobic parts of the membrane protein are covered by its hydrophobic side, while the hydrophilic face of the peptides remains exposed to the solvent. But this process is likely to be stochastic in the sense that there remain hydrophobic gaps on the membrane protein surface, as well as on some of the not perfectly placed NSP peptides. These remaining hydrophobic surfaces would then serve as nucleation points for protein aggregation. Remaining detergent molecules originating from the solubilization and purification of the membrane protein may shield these remaining hydrophobic surfaces.

This reviewer raises very important and fundamental questions, and the exact same questions apply to the nanodisc and SMA polymer systems as well. From our experiments, detergent appears to be completely eliminated from the peptidisc, as characteristic effects of a detergent environment on protein activity and stability are not seen once the protein is transferred into the peptidisc. The experimental data show that aggregation is not occurring and the system are very stable in water solution. We have also revised the Discussion to clarify how the peptide may possibly orientate itself around the membrane protein in a peptidisc to account for the issue of exposed alkyl chains.

3) Related to the above comment: Do peptidiscs still contain detergents? It is well known that quantitative detergent exchange is not a trivial endeavor. Often the initial detergent used to solubilize the membrane protein remains present to some degree. The authors need to show, whether in the peptidiscs there are still some detergents remaining (next to annular lipids and the NSP peptide).

From the high degree of correlation between our calculated and observed masses for the peptidiscs, it is likely that most, if not all of the detergent is removed from the peptidisc. Furthermore, in all cases of exchange from detergent to peptidisc, significant changes in enzyme activity or thermostability are observed, suggesting that even if a small amount of detergent remains, the denaturing effect of a detergent environment does not. We have discussed this point in the expanded text.

4) How big is the NSP belt compared to detergent belts of different sizes? This question is highly relevant for membrane protein crystallization. The peptidisc appears more compact than a nanodisc and membrane proteins in peptidiscs might be crystallized. Did the authors try to crystallize their five membrane proteins used in this study, for which high resolution crystal structures (determined in detergent) exist?

We are also very excited with the perspective of getting crystals! Crystal trials are ongoing, and based on EM comparisons of MalFGK_2_, we also think the peptidisc is more compact than the nanodisc. However, the crystal optimization and use of the peptidisc for crystallization studies are outside the scope of this current paper.

5) Does the peptidisc really mimic the natural environment of a membrane protein? The authors carefully worked out that the peptidisc contains annular lipids. However, annular lipids are also contained in detergent-purified samples. The authors should address this question by comparing the content of annular lipids side-by-side and over time using the same transporter purified in detergent as well as in peptidiscs and measuring by mass spectrometry the mass (or loss of mass over time) of the entire complex.

This question applies to all other membrane scaffold systems as well. The peptidisc is clearly more gentle environment than the detergent micelle. Comparison of BRC reconstituted in SMA, nanodiscs, peptidiscs, and proteoliposomes show that all these membrane mimetics lead to a comparable increase in the protein’s thermostability. However, the peptidisc also preserves protein ability to bind surface ligands as shown in the manuscript with FhuA/ColM/Ferricrocin and MalFGK_2_-MalE. In detergents, these proteins show binding/activity that is inconsistent with their performance in a lipid system.

6) The manuscript lacks a proper Discussion. Most problematically, the authors do not discuss at all that the concept of peptides used as detergent surrogates is not novel at all. Rather, it was first described in the 80s and 90s as peptitergent (you can find this word even on Wikipedia). See also in Schafmeister CE, Miercke LJ, Stroud RM. Structure at 2.5 A of a designed peptide that maintains solubility of membrane proteins. Science. 1993 Oct 29;262(5134):734-8. In addition, there are lipopeptide detergents (LPDs), nano-structured β-sheets (see Tao H, Lee SC, Moeller A, Roy RS, Siu FY, Zimmermann J, Stevens RC, Potter CS, Carragher B, Zhang Q. Engineered nanostructured β-sheet peptides protect membrane proteins. Nat Methods. 2013 Aug;10(8):759-61). Further, there was the development of short peptides for membrane stabilization, which are much cheaper to produce than the 37 aa NSP peptide (see Kiley P, Zhao X, Vaughn M, Baldo MA, Bruce BD, Zhang S. Self-assembling peptide detergents stabilize isolated photosystem I on a dry surface for an extended time. PLoS Biol. 2005 Jul;3(7):e230. Epub 2005 Jun 21.)Important is the question whether at all or to what extent the NSP peptide is superior to these previously described methods using amphiphilic peptides, in particular the short designer peptides as described in Kiley et al., 2005). Critically, this point needs to be addressed experimentally by a functional assay including the maltose transporter and/or the BRC.

We absolutely agree with the reviewer. The manuscript was initially submitted as a Short Communication, which has text limitations. We have now rewritten sections Introduction and Discussion so that many of the well thought out comments above can be addressed. In the revised manuscript, we explain why the peptidisc is superior than previous peptide-based methods, as well as addressing synthetic polymers protein scaffold based systems. We comment on possible orientations of the peptide in the peptidisc, as well as the possibility of detergents remaining in the particles.

*The Discussion should also include the concept of membrane protein stabilization via peptide-protein interactions (peptidisc) versus lipid protein interactions (nanodisc/saposins/proteoliposomes). And finally, it needs to address the amphipols (which are conceptually most closely linked to peptitergents and peptidiscs) in more detail.*

We have modified the Discussion to encompass these points in more detail.

[Editors’ note: the author responses to the re-review follow.]

The manuscript has been improved but there are some remaining issues that need to be addressed before acceptance, as outlined below:1) What is interesting in the revised version is the fact that the sequence of the original NSP peptide as it is present in ApoA1 was in fact reversed and that this resulted in an increase of solubility of the peptide. Why was this information completely lacking in the first version?

The sequence utilized through this study has been consistently reported in the supplementary table and labeled generically as NSP (i.e. nano-scaffold peptide) because we did not find many differences in reconstitution performance compared to the NSP sequence described by Kariyazono et al. However, issues of lower solubility became more evident to us after submission of the first manuscript version. We therefore felt it important to present this information in the revision, include additional experimental data and also to determine the hydrophobicity moment to explain for this difference (Figure 8—figure supplement 1). To prevent confusion with Kariyazono et al., we had to rename this sequence to NSP_r_ in the revised version, and it was an obvious mistake in hindsight not to label it as such in the first submission.

2) The exact sequence of the NSPr peptide should be provided directly in the Materials and methods section under section "Biological reagents and peptides" (I had difficulties finding it in Supplementary file 1).

The sequence of NSPr has been added to the manuscript Materials and methods, section entitled “Peptides”, in addition to Supplementary file 1.

3) The authors are now stating in the manuscript that the method is inexpensive. However, I would still be interested to know how much a mg or a gram costs and how much you will need for an experiment (by providing a reasonable range).

As an example, the peptide content of MsbA-peptidisc represents ¼ of the total molecular weight. Therefore the reconstitution of MsbA in peptidisc using the on-bead method in quantities suitable for crystallization screening (~20mg/mL, 200µL) utilizes at least of ~1mg peptide. The cost of the peptide can be as low as ~5-15$/mg, pricing dependent on supplier, quantity and purity, thus the peptide cost for crystallization screen can be ball parked in the ~5-15$ range. An equivalent amount of MSP (https://www.sigmaaldrich.com/catalog/product/sigma/m6574?lang=en&region=CA) would cost close to $80. There is, of course, additional peptide lost during purification steps (IMAC, SEC, concentration steps). The final cost in our lab to prepare this amount of MsbA in peptidisc for crystal screening is $20, assuming we lose about 60% of the MsbA during the purification steps. In comparison, each Hampton sitting drop crystal plate will cost close to 5$. In the case of EM experiments, the cost is even lower since less protein is required (5-10 mg/mL). In either case, the on-bead method is important to limit the total cost because the peptidisc solution can be re-utilized multiple times, as only the requisite amount of peptide binds the protein. In comparison, the on-column reconstitution is more wasteful because the initial 10 fold excess of peptide cannot be recovered as it fractionates with small size impurities. The on-gel reconstitutions utilize approximately 2-8µg of NSP_r_, so the cost of optimization is extremely low. Another very important consideration is labor time generally associated with the preparation of other membrane scaffolds. This is not the case with the peptidisc since the product can be manufactured.

4) Concerning the impurities of the peptide, the authors gave an interesting answer in the response letter (i.e. that the great majority of the impurity is in fact the same peptide missing its final amino acid). This should be mentioned in the manuscript as well.

This information and the degree of peptide purity are now presented together in the same place in the Materials and methods, section entitled “Peptides”.

5) The authors refer to a homepage to order/obtain the peptide: www.peptidisc.com. However, the homepage does work yet. It seems that the authors will or have already founded a start-up company to commercialize the peptidisc. Potential financial competing interests have to be declared.

We recently started a website with the goal to deliver NSP_r_ in a kit along model membrane proteins so that other researchers can easily reproduce the core reconstitution experiments we present in this manuscript before proceeding on their own target. We mentioned in the manuscript “To aid accessibility to the academic community, bulk NSP_r_ peptides and core protocols are available at www.peptidisc.com”, and we can declare a competing financial interest where necessary. Our goal is to make the method accessible to all and the anticipated cost will reflect a small revenue to cover its distribution and efforts to ensure its proper application including technical advices. We had not released the website yet due to the paper remaining under review and will do soon after publication. We have also named the other various source of peptide manufacturers employed in this study. The peptidisc presents an additional interesting opportunity in comparison to other scaffolds, as its use is not protected by intellectual property so it can be easily utilized by academics and industry alike.